# Structural basis for transthiolation intermediates in the ubiquitin pathway

Tomasz Kochańczyk[1,2], Zachary S. Hann[1,3], Michaelyn C. Lux[3,4], Avelyn Mae V. Delos Reyes[4,5], Cheng Ji[4], Derek S. Tan[3,4,5 ✉] & Christopher D. Lima[1,2,3 ✉]

Transthiolation (also known as transthioesterification) reactions are used in the biosynthesis of acetyl coenzyme A, fatty acids and polyketides, and for post-translational modification by ubiquitin (Ub) and ubiquitin-like (Ubl) proteins[1–3]. For the Ub pathway, E1 enzymes catalyse transthiolation from an E1–Ub thioester to an E2–Ub thioester. Transthiolation is also required for transfer of Ub from an E2–Ub thioester to HECT (homologous to E6AP C terminus) and RBR (ring-between-ring) E3 ligases to form E3–Ub thioesters[4–6]. How isoenergetic transfer of thioester bonds is driven forward by enzymes in the Ub pathway remains unclear. Here we isolate mimics of transient transthiolation intermediates for E1–Ub(T)–E2 and E2–Ub(T)–E3^HECT complexes (where T denotes Ub in a thioester or Ub undergoing transthiolation) using a chemical strategy with native enzymes and near-native Ub to capture and visualize a continuum of structures determined by single-particle cryo-electron microscopy. These structures and accompanying biochemical experiments illuminate conformational changes in Ub, E1, E2 and E3 that are coordinated with the chemical reactions to facilitate directional transfer of Ub from each enzyme to the next.

Ubiquitin and Ubl proteins are conjugated to protein or lipid targets to regulate target activity, stability, localization or interactions[3,7–9]. Conjugation is achieved through the cascade of activities of dedicated E1 activating enzymes, E2 conjugating enzymes and E3 protein ligases that catalyse modular synthesis of diverse Ub and Ubl products with distinct topologies or composition from a limited number of Ub or Ubl building blocks[10]. Transthiolation reactions are central to Ub and Ubl conjugation pathways because they are used for transfer of thioester-activated Ub or Ubl from E1 to all E2 proteins, and from some E2 proteins to specialized E3 proteins. Conjugation reactions typically result in formation of an isopeptide bond between the C-terminal glycine of the Ub or Ubl and a lysine side chain of the protein target, although alternative substrates and linkages are possible[11].

Uba1, the E1 activating enzyme for Ub, and related Ubl E1 enzymes utilize a multi-domain architecture and conformational changes termed domain alternation to create distinct chemical environments that promote adenylation, thioester bond formation or transthiolation[12–14]. E1 binding of ATP•Mg$^{2+}$ and Ub is followed by adenylation of the Ub C terminus to generate Ub-AMP and inorganic pyrophosphate (PP$_i$•Mg$^{2+}$) (Fig. 1a). Subsequent thioester formation between the E1 catalytic cysteine and Ub C-terminal glycine is rate-limiting because it requires release of PP$_i$•Mg$^{2+}$ and 130° rotation of the E1 catalytic cysteine domain (SCCH) to replace amino acid residues in the adenylation active site with those required for thioesterification[14,15]. After E1–Ub(T) formation (where ~ denotes a thioester) and AMP release, the E1 SCCH domain is driven back to its original conformation for transthiolation to E2 by remodelling of the adenylation active site through binding of ATP•Mg$^{2+}$ and a second Ub[16,17] (Ub(A); A denotes Ub in the adenylation site).

This doubly loaded E1 is most adept at recruiting individual E2 enzymes (there are 35 in human) to catalyse transthiolation of E1–Ub(T) to an E2 active site cysteine[18–20] to generate E2–Ub(T). Transthiolation from E1 to E2 represents another rate-limiting step. Whereas singly loaded E1 lacking Ub(A) or ATP•Mg$^{2+}$ can transfer Ub(T) to E2, occupancy at the adenylation site by either ATP•Mg$^{2+}$ or Ub-AMP enhances the transfer rate by tenfold[20]. These ligands differ in composition and contacts with E1, especially those involving E1 side chains that participate in adenylation and domain alternation, so it remains unclear how they enhance or synergize with transfer.

Transthiolation is required for transfer of Ub(T) from E2–Ub(T) to a catalytic cysteine in HECT and RBR E3 proteins to form an E3–Ub(T) thioester en route to Ub(T) conjugation to targets[4] (Fig. 1a). This mechanism is distinct from that of scaffolding RING and U-box E3 enzymes that mediate Ub(T) conjugation through non-covalent activation of E2–Ub(T)[21,22]. HECT E3 enzymes (there are 28 in human) include a conserved HECT domain with an N-terminal lobe (N-lobe) and a C-terminal lobe (C-lobe), with the C-lobe containing the catalytic cysteine[23]. Analogous to E1 domain alternation, the C-lobe rotates 150° between conformations that catalyse transthiolation from E2–Ub(T) to E3 or Ub conjugation from E3–Ub(T) to targets. HECT E3s exhibit specificity for a unique class of E2s that are adept at transthiolation, as exemplified by UbcH7 and UbcH5 family members[24]. Existing structures of E2–Ub(T)–E3^HECT transthiolation mimics reveal Ub(T) coordinated by the HECT C-lobe far from E2, in conformations similar to that observed for E3^HECT–Ub(T) products in the presence or absence of E2[25–27]. Thus, earlier intermediate conformations that precede the product state have not yet been characterized.

[1]Structural Biology Program, Sloan Kettering Institute, New York, NY, USA. [2]Howard Hughes Medical Institute, New York, NY, USA. [3]Tri-Institutional PhD Program in Chemical Biology, Memorial Sloan Kettering Cancer Center, New York, NY, USA. [4]Chemical Biology Program, Sloan Kettering Institute, New York, NY, USA. [5]Pharmacology Graduate Program, Weill Cornell Graduate School of Medical Sciences, Memorial Sloan Kettering Cancer Center, New York, NY, USA. ✉e-mail: tand@mskcc.org; limac@mskcc.org

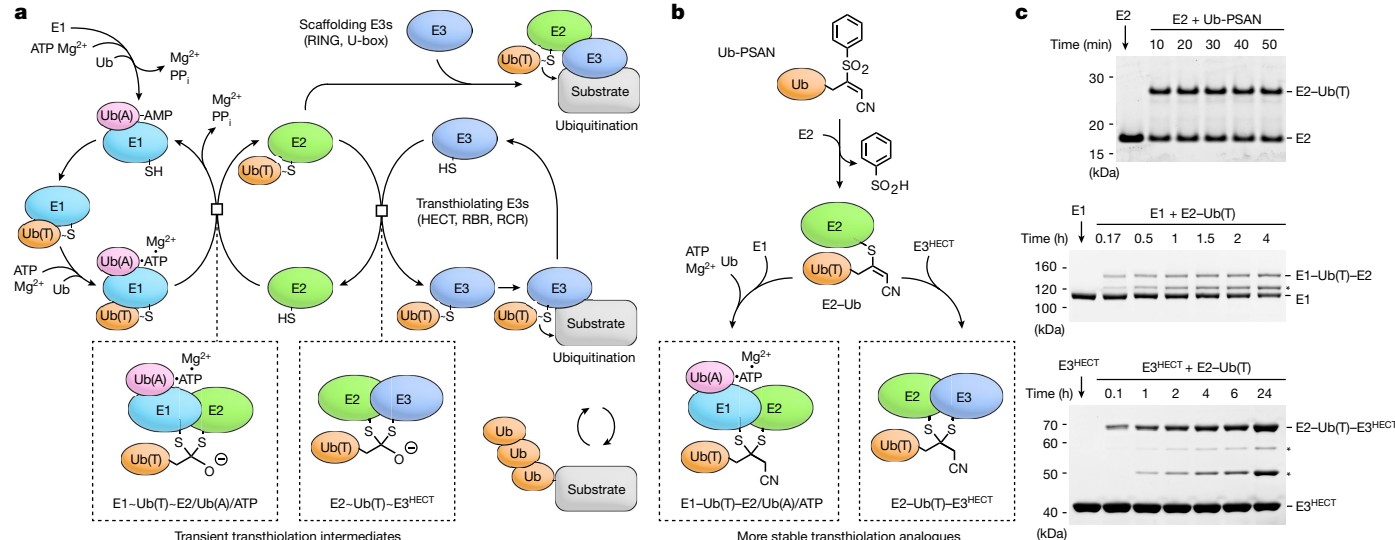

**Fig. 1 | The Ub pathway, transthiolation intermediates and strategy to obtain stable mimics. a**, Reactions in the Ub-conjugation cascade. **b**, Reaction of the Ub-PSAN probe with E2 to form E2–Ub vinyl thioether and subsequent reactions with E1 and E3[HECT] to obtain stable analogues of transthiolation intermediates. **c**, SDS–PAGE analysis of reactions of Ub-PSAN with E2 (top) and E2–Ub(T) vinyl thioether with E1 (middle) and E3[HECT] (bottom). Asterisks indicate bands for side- or retro-reaction products (Extended Data Fig. 1d,e and Supplementary Fig. 1).

Structural investigation of transthiolation intermediates is challenging because they are transient and unstable. Further, mechanisms that impose directionality during otherwise isoenergetic transthiolation reactions remain unclear. Here we use a chemical approach to form more stable analogues of transient transthiolation intermediates between native Ub enzymes, including E1, E2 and E3[HECT] enzymes, and near-native Ub[28]. This strategy enabled capture and visualization of a continuum of cryo-EM structures for transthiolation intermediates of E1–Ub(T)–E2 and E2–Ub(T)–E3[HECT] that define chemical steps and conformational transitions that promote transthiolation and Ub transfer.

## Stable analogues of transthiolation intermediates

We developed a selective bis-electrophilic 3-[phenylsulfonyl]–4-aminobut-2-enenitrile (PSAN) probe to investigate transthiolation by trapping two cysteines covalently while mimicking a tetrahedral intermediate at the Ub C terminus[28] (Fig. 1b and Extended Data Fig. 1a–c). Aminolysis of Ub(–2) acyl azide (Ub residues 1–74) with H$_2$N-Gly–PSAN provided Ub-PSAN. This construct was combined with *Schizosaccharomyces pombe* E2 Ubc4, which was selected because it is the homologue of human UbcH7 and UbcH5, E2 proteins that are specialized for transthiolation to HECT or RBR E3 ligases[24]. The predominant product was a crosslink between Ub-PSAN and the E2 active site cysteine (Cys85), and serine substitution of surface-exposed Cys21 and Cys107–modifications that do not alter E2 activity–maximized formation of the desired product[17] (Fig. 1c and Supplementary Fig. 1). The resulting E2–Ub vinyl thioether adduct was combined with Uba1 (E1) or Pub2[HECT] (a homologue of the NEDD4 family of E3[HECT] enzymes) to yield atomically tailored dithioacetal analogues of E1–Ub(T)–E2 and E2–Ub(T)–E3[HECT] transthiolation intermediates with substitution of the Ub(T) C-terminal Gly76 oxyanion by a cyanomethyl group (Fig. 1b,c and Supplementary Fig. 1). This approach maintains a single carbon between the cysteine thiols, as in the native tetrahedral intermediates, while preserving native side chains of Ub and residues that comprise E1, E2 and E3 active sites. Purified E1–Ub(T)–E2 and E2–Ub(T)–E3[HECT] reverted slowly to enzyme–Ub(T) adducts or to unmodified enzymes in a manner dependent on time, pH and temperature, perhaps congruent with the reversible and labile characteristics of native transthiolation intermediates (Extended Data Fig. 1d,e). Nonetheless, the E1–Ub(T)–E2 and

E2–Ub(T)–E3[HECT] conjugates remained stable long enough for analysis by single-particle cryo-EM.

## Structures of E1–Ub(T)–E2 transthiolation analogues

To obtain doubly and singly loaded E1–Ub(T)–E2 complexes with and without Ub(A) in the same sample, E1–Ub(T)–E2 was incubated with substoichiometric Ub (approximately 0.8 equivalents) and excess ATP•Mg$^{2+}$. Samples were vitrified and analysed using cryo-EM for doubly and singly loaded complexes to obtain reconstructions at overall resolutions of 2.5 and 2.8 Å, respectively (Extended Data Fig. 2 and Supplementary Table 1). Densities for E1 and E2 active site thiols and Ub(T) Gly76 are apparent and consistent with covalent linkage at a single dithioacetal carbon (Fig. 1b and Extended Data Fig. 3a,b). The cyanomethyl group of the linker, which replaces the negatively charged oxygen of the tetrahedral transthiolation intermediate, is positioned next to the N-terminal end of helix 18 of E1 that includes the E1 active site cysteine (Fig. 2a and Extended Data Fig. 3a,b). Positive electrostatic potential of the helix dipole or hydrogen bonding may stabilize the negative electrostatic potential of the oxyanion intermediate.

Heterogeneity of Ub(T) was resolved by focused 3D classification to yield 10 reconstructions for doubly loaded complexes at overall resolutions of 2.8 Å and for singly loaded complexes at overall resolutions of 3.2 Å. Although the local resolution for Ub is lower, it is sufficient to position Ub to reveal a conformational continuum with Ub(T) moving from a position next to the first catalytic cysteine half-domain (FCCH) of E1 to a position adjacent to the E2 (Fig. 2b, Extended Data Figs. 3c and 4 and Supplementary Fig. 2). E1 donates Ub(T) to E2 during transthiolation, so we define Ub(T) positions proximal to E1 as Ub(T)-donor, those proximal to E2 as Ub(T)-acceptor, and positions in between as Ub(T)-intermediate. The C terminus of Ub is covalently linked to the E1 and E2 active site cysteines, whereas the body of Ub(T) transits between donor and acceptor conformations through a 180° rotation and 35 Å translation as measured from the Ub N-terminus (Fig. 2c–e). The Ub(T) position (state 1, overall resolution 2.8 Å) proximal to FCCH resembles that observed in a crystal structure of doubly loaded E1 in the absence of E2[29] (Extended Data Fig. 3d). The Ub(T) position most proximal to E2 (state 10, overall resolution 2.81 Å) differs by rotation of 55° from that reported in a crystal structure in which the E1 and E2 active site

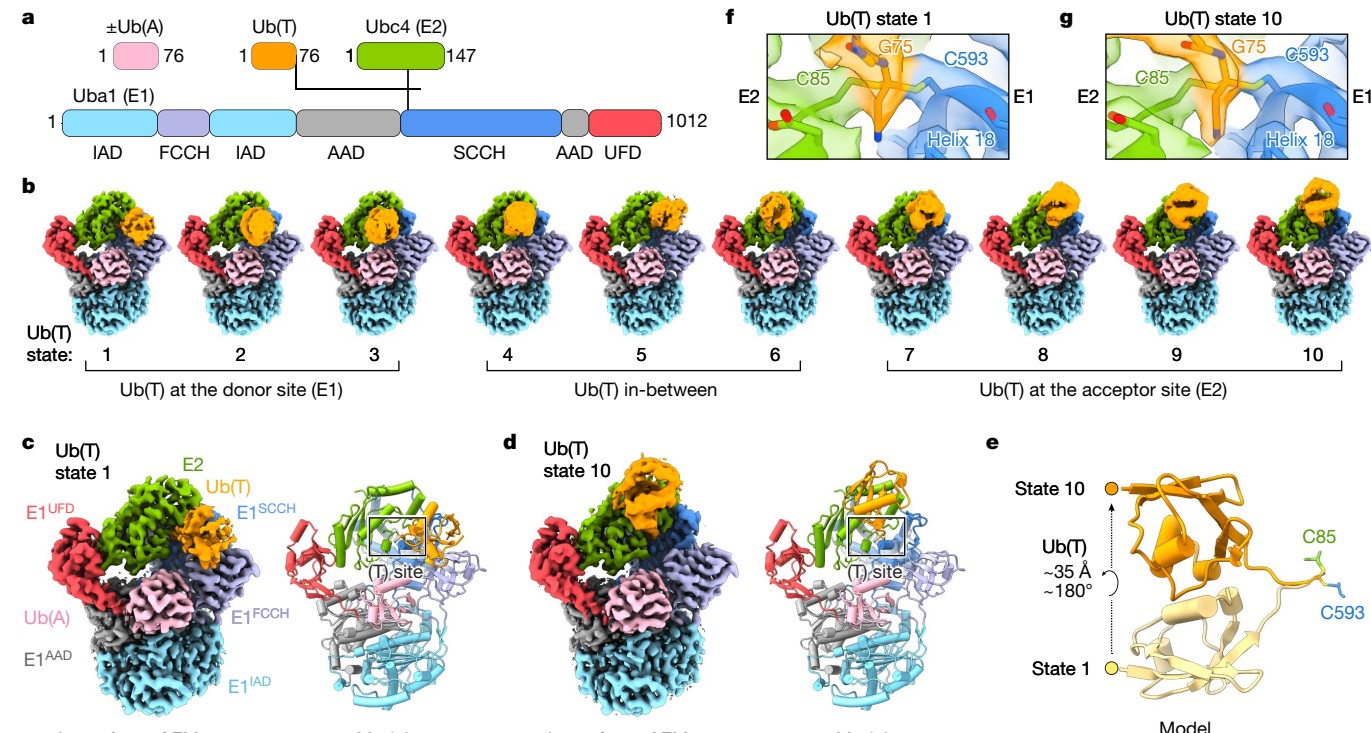

**Fig. 2 | Reconstructions of doubly loaded E1 reveal Ub(T) conformational changes and the transthiolation active site. a**, Schematic of Ub (Ub(A) and Ub(T)), E2 (Ubc4) and E1 (Uba1) with colour-coded domains and the crosslink indicated by lines between Ub(T), E2 and E1. UFD, C-terminal Ub fold domain. **b**, Ten reconstructions of doubly loaded E1 obtained by 3D classification showing different conformations of Ub(T) in states 1–10 from its donor to acceptor positions. **c,d**, Reconstructions of states 1 (**c**) and 10 (**d**) shown next to models highlighting movement of Ub(T). Electron microscopy (EM) densities and models are colour-coded as in **a**. **e**, Cartoon representation of models for Ub(T), illustrating the 180° rotation and 35 Å translation between states 1 and 10. **f,g**, Magnified views of the transthiolation site (T-site) with overlaid electron microscopy densities and model for doubly loaded E1 state 1 (**f**) and state 10 (**g**), centred on the cyanomethyldithioacetal mimic of the E1–Ub(T)–E2 tetrahedral intermediate. Isosurface levels contoured at 0.5 (**b**), 0.63 (**c,d**), 0.54 (**f**) and 0.60 (**g**).

cysteines were disulfide-bonded with Ub linked to E2 Cdc34 through a lysine adjacent to the active site cysteine[30] (Extended Data Fig. 3e). The conformation observed for Ub in E1–Cdc34–Ub is similar to that observed for Cdc34–Ub in the absence of E1[31,32].

We speculated that remodelling of E1 and E2 active site residues could impart directionality during transthiolation, but amino acids and side-chain conformations around the transthiolation active site remain relatively constant throughout conformational changes for Ub(T) (Fig. 2f,g). Thus, to gain further insight we next explored conformational dynamics within the E1–E2 domains of the complex.

## Transthiolation and adenylation are coupled for E1

Doubly loaded E1 bound to Ub(A) and ATP•Mg²⁺ is most adept at Ub(T) transfer to E2 but prior biochemical studies have shown that ATP•Mg²⁺ alone or the Ub(A)-AMP adenylate can also enhance Ub(T) transfer from E1-Ub(T)[20]. To determine a structural basis for these observations, particles for doubly and singly loaded complexes were individually subjected to 3D-variablity analysis focused on E1–E2 to generate 5 clusters with reconstructions at overall resolutions ranging from 2.78 to 2.86 Å for doubly loaded E1 and 2.95 to 3.16 Å for singly loaded E1 (Extended Data Figs. 4 and 5 and Supplementary Table 1). These reconstructions capture a conformational continuum for E1 involving up to 8° rotation of the SCCH domain relative to E1 adenylation domains (inactive adenylation domain (IAD) and active adenylation domain (AAD)) and up to 8 Å translation measured at its distal regions (Fig. 3a,b, Extended Data Figs. 4 and 5 and Supplementary Fig. 2). Movements were evident for doubly and singly loaded complexes, but their trajectories differed, as did conformations of E2 and Ub, domains adjacent to the E1 SCCH

domain, and residues that encompass the E1 active site for adenylation (Fig. 3a,b, Extended Data Fig. 5 and Supplementary Video 1).

To determine whether E1 domain movements from cluster 1 to cluster 5 were correlated with Ub(T) movements from donor to acceptor positions for singly loaded complexes bound to ATP•Mg²⁺ (Fig. 3c and Extended Data Fig. 5i), we used 3D classification to isolate 10 states for Ub(T) for each of the five E1 clusters to generate 50 reconstructions. Despite the limited number of particles per class, each is at sufficient resolution to observe densities for ATP•Mg²⁺, secondary structure and side chains throughout E1 and E2 (Extended Data Figs. 4 and 6 and Supplementary Fig. 2). Local resolution was lower for Ub(T), but densities were sufficient to establish unique positioning for Ub(T) in each reconstruction. These reconstructions show that E1 SCCH and E2 rotations are not correlated with changes in the distribution between Ub(T) in Ub(T)-donor and Ub(T)-acceptor positions (Fig. 3d). Instead, these changes are correlated with movement of the E1 FCCH domain toward the empty Ub(A) binding site (Fig. 3a and Extended Data Fig. 5i). The Ub(T) distribution is biased toward the donor position, perhaps because the FCCH domain moves and maintains interactions with Ub(T) in the donor position. Although Ub(T) transfer appears suboptimal, Ub(T) occupies the acceptor position in 20–25% of the states within each cluster, suggesting that binding of ATP•Mg²⁺ and E1 contacts to Mg²⁺ and ATP β and γ phosphates might stimulate transfer by positioning the SCCH domain in an upright configuration for Ub(T) transfer to E2[17,20].

Doubly loaded complexes represent those that are most adept at Ub(T) transfer. 3D classification of ten states within each of its five clusters generated reconstructions at resolutions suitable to visualize and model ATP•Mg²⁺, the Ub(A)-adenylate, pyrophosphate and all protein components including Ub(T), although Ub(T) is the most

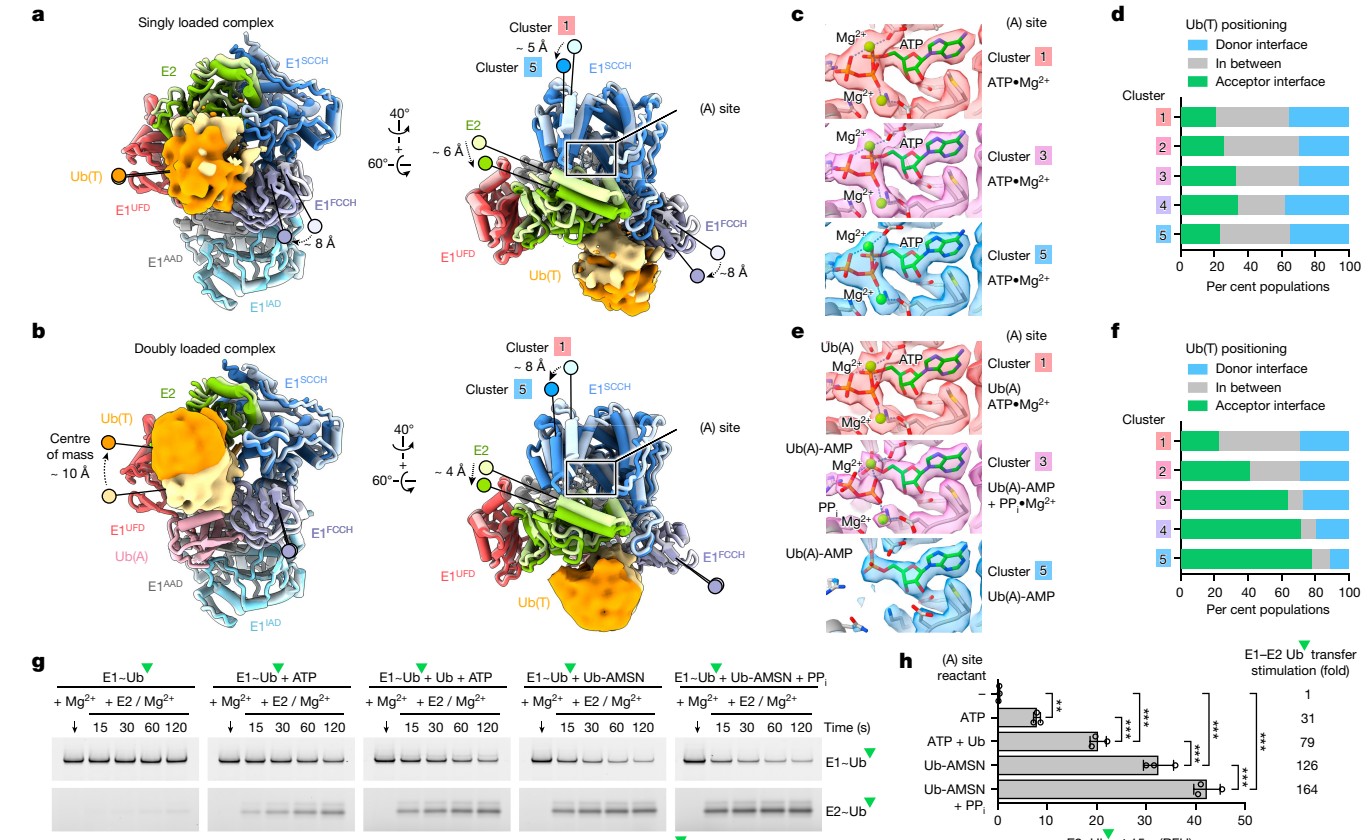

**Fig. 3 | Conformational changes for E1–Ub(T)–E2 complexes and coupling of Ub(A) adenylation and Ub(T) transitions to E2 for doubly loaded E1–Ub(T)–E2. a**, Superposed models of clusters 1 and 5 for singly loaded E1 with Ub(T) in solid as isosurface rendering of 5 Å lowpass-filtered EM densities showing translation of E1 FCCH and SCCH domains and E2 as indicated by labels and arrows; rotational changes are not labeled. **b**, Superposed models of cluster 1 and 5 for doubly loaded E1 rendered as in **a**, highlighting translation of the E1 SCCH domain, E2 and Ub(T). **c,d**, Electron microscopy densities and models for the adenylation (A) active site (**c**) and Ub(T) conformations (%) per E1 cluster (**d**) (Extended Data Fig. 6) indicate that E1 and E2 rotations do not correlate with changes in Ub(T) positioning. **e,f**, Electron microscopy densities for the adenylation (A) active site (**e**) and Ub(T) conformations (%) per E1 cluster (**f**)

(Extended Data Fig. 7) indicate that E1 and E2 rotations correlate with loss of PP$_i$ in the adenylation site and movement of Ub(T) from donor to acceptor positions. Isosurface levels contoured at 0.23–0.24 (**a,b**); 1.06, 1.01 and 0.9 for clusters 1, 3 and 5 (**c**); and 1.19, 1.2 and 1.15 for clusters 1, 3 and 5 (**e**). Ub(T) conformations (%) per E1 cluster are plotted as a bar graph (Extended Data Fig. 7). **g**, Stimulation of E1-to-E2 transthiolation by E1 adenylation site ligands. SDS–PAGE gels indicating E1-Ub(T) and E2-Ub(T) over time after initiating reactions with E2. **h**, Quantification of transthiolation rates from **g** (Extended Data Fig. 5k and Supplementary Fig. 3). RFU, relative fluorescence units. Green triangle indicates fluorescein. Bars represent mean ± s.d of $n$ = 3 replicates. Statistical differences by two-sided one-way ANOVA with Tukey's test, ***$P$ < 0.001, **$P$ < 0.01.

mobile element in these structures (Extended Data Figs. 4 and 7 and Supplementary Fig. 2). These reconstructions reveal coordination among E1 conformational changes, Ub(T) transfer and Ub(A) adenylation (Fig. 3b,e,f and Extended Data Fig. 5j). First, rotation of the SCCH domain and E2 is accompanied by adenylation of Ub(A) molecule. In the first two clusters, ATP•Mg²⁺ is evident as are E1 contacts to the β and γ phosphates. In clusters 3 and 4, adenylation is evident as are E1 contacts to PP$_i$•Mg²⁺. In the last cluster, PP$_i$•Mg²⁺ is absent and there is loss of densities for half of the adenylation active site. Second, SCCH and E2 rotations and Ub(A) adenylation are correlated with transition of Ub(T) from donor to acceptor positions with 20% of Ub(T) in acceptor positions in cluster 1 to nearly 80% in cluster 5 (Fig. 3f and Supplementary Video 1).

Analysis of doubly loaded E1 reconstructions suggests that Ub(A) adenylation and PP$_i$•Mg²⁺ release are coupled with transfer of Ub(T) to E2[20,33] and that singly loaded complexes bound to ATP•Mg²⁺ are suboptimal for transfer. To test this, we purified E1-Ub(T) and measured the rate of Ub(T) transfer to E2 in single-turnover reactions in the presence of Mg²⁺, or with Mg²⁺ combined with ATP (analogous to singly loaded complexes), Ub and ATP (analogous to clusters 1 and 2 for doubly loaded complexes), the non-hydrolysable Ub-AMP mimic Ub-AMSN[15,34]

and PP$_i$ to bypass adenylation (analogous to clusters 3 and 4 for doubly loaded complexes), and Ub-AMSN alone (analogous to cluster 5 for doubly loaded complexes) (Fig. 3g,h, Extended Data Fig. 5k and Supplementary Fig. 3). Addition of ATP•Mg²⁺ or Ub-AMSN stimulated rates of E1–E2 Ub(T) transfer by 31-fold and 126-fold, respectively, based on the amount of Ub(T) transferred in the first 15 s. Notably, addition of Ub(A) and ATP•Mg²⁺ stimulated the rate by 79-fold, faster than ATP•Mg²⁺ but slower compared to Ub-AMSN, consistent with adenylation or PP$_i$•Mg²⁺ release as a rate-limiting step. However, reactions with PP$_i$•Mg²⁺ and Ub-AMSN, which bypasses the adenylation step, increased the rate by 164-fold, the highest observed. So, although adenylation and PP$_i$•Mg²⁺ release may be rate-limiting, interactions with these ligands result in maximal transthiolation activity.

Analysis of doubly loaded E1 reconstructions provides a structural basis for why transfer from E1-Ub(T) to E2 with Ub-AMSN/PP$_i$•Mg²⁺ is more efficient relative to Ub-AMSN, ATP•Mg²⁺, or even Ub(A) with ATP•Mg²⁺. First, the complex with Ub-AMSN/PP$_i$•Mg²⁺ provides contacts in the adenylation active site between E1 and PP$_i$•Mg²⁺ that favour an upright SCCH conformation that is optimal for transthiolation to E2 while bypassing adenylation and preventing the reverse reaction. Second, the occupied Ub(A) site and contacts between Ub(A) and the

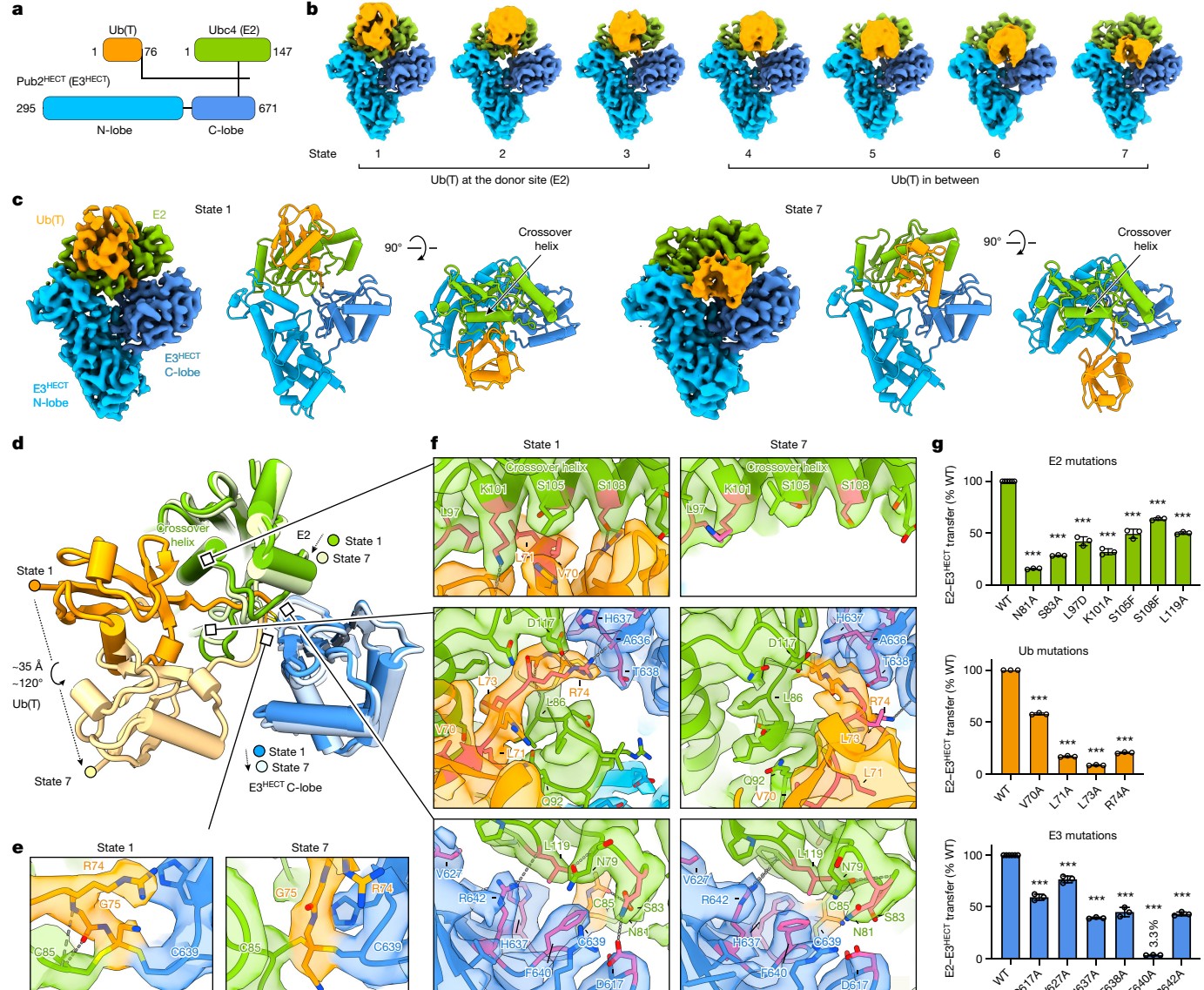

**Fig. 4 | E2–Ub(T)–E3^HECT structures reveal active site remodelling and conformational changes for E2 and Ub(T). a**, Schematic of Ub (Ub(T)), E2 (Ubc4) and E3^HECT (Pub2) with domains colour-coded and the crosslink indicated by lines between Ub(T), E2 and E3. **b**, Seven reconstructions obtained for E2–Ub(T)–E3^HECT showing conformations of Ub(T) in states 1–7 from donor to intermediate positions (unsharpened maps). **c**, Reconstructions and models of state 1 and state 7 next to orthogonal views (sharpened maps), highlighting movement of Ub(T). **d**, Superposed models showing movement of Ub(T) between states 1 and 7. **e**, Magnified views of transthiolation sites (T-site) with electron microscopy densities for states 1 and 7, showing the cyanomethyldithioacetal mimic of the E2–Ub(T)–E3^HECT tetrahedral intermediate and Ub(T) Arg74

side chain. **f**, Magnified views of areas of interest highlighting differences in E2 contacts with Ub(T) between states 1 and 7 with residues labelled. **g**, Results of transthiolation assays with E2, Ub and E3 mutations to probe putative protein– protein interactions during transthiolation, quantified after normalization to wild type (WT) with indicated mutations labelled and colour-coded. Densities and models colour-coded according to **a**. Isosurface levels contoured at 0.47– 0.52 (**b**); at 0.72, 0.75 and 0.71 for state 1 and 0.5, 0.6 and 0.59 for state 7 (**c,e,f**). Bars represent mean ± s.d of *n* = 3 replicates (*n* = 9 and *n* = 6 for wild type in E2 and E3 mutations sets, respectively). Statistical differences between wild type and mutants by two-tailed unpaired *t*-test, ***P < 0.001 (Supplementary Fig. 4).

FCCH domain restrict FCCH domain movements to enable Ub(T) release from the E1 and transfer to E2 (Supplementary Video 1). These observations suggest that Ub(A) adenylation and PP$_i$•Mg$^{2+}$ release co-occur and are coupled with transthiolation. As these changes are correlated, it is equally plausible that transthiolation promotes Ub(A) adenylation and PP$_i$•Mg$^{2+}$ release.

## Structures of E2–Ub(T)–E3 transthiolation analogues

E2-Ub(T) is the donor for Ub(T) in ATP-independent E2–E3 transthiolation reactions. To determine structural features that underlie

directionality in this process, E2–Ub(T)–E3^HECT (Fig. 4a) was vitrified and analysed using single-particle cryo-EM to yield a reconstruction with an overall resolution of 3.1 Å (Extended Data Fig. 8 and Supplementary Table 1). Three-dimensional variability analysis was used to isolate 7 reconstructions with overall resolutions ranging from 3.2 to 3.5 Å (Fig. 4b, Extended Data Figs. 4 and 8 and Supplementary Fig. 2). These structures reveal a conformational continuum with Ub(T) transitioning from a position proximal to E2 for state 1 (3.2 Å overall resolution) to a position between the E2 and the C-lobe of E3^HECT in state 7 (3.3 Å overall resolution) while the Ub(T) C terminus remains covalently linked to the E2 and E3^HECT catalytic cysteine residues (Fig. 4c, Extended Data Fig. 8

and Supplementary Video 2). Densities and distances between the E2 and E3[HECT] active site thiols and Ub(T) Gly76 are consistent with linkage to a single dithioacetal carbon (Fig. 4d,e), as observed for E1–Ub(T)–E2, although densities between the E2 active site thiol and Ub(T) Gly76 are weak for states 2 and 3 (Extended Data Fig. 8). The cyanomethyl group of the linker, which replaces the negatively charged oxygen of the tetrahedral transthiolation intermediate, is positioned proximal to the Arg74 side-chain guanidinium of Ub(T) in state 1 (Fig. 4e). Arg74 is highly conserved in Ub, and its positive electrostatic potential or hydrogen bonding capacity may stabilize the negative electrostatic potential of the oxyanion intermediate. E2 donates Ub(T) to E3[HECT], so we define Ub(T) positions proximal to E2 as Ub(T)-donor and positions as Ub(T) transitions toward the C-lobe of E3[HECT] as Ub(T)-intermediate.

The orientations of E3[HECT] N- and C-lobes remain mostly unchanged through all states, closely matching prior structures with the C-lobe in a transthiolation-competent architecture in the inverted T configuration with the 'Tyr-lock' interface—previously shown to be important for transthiolation[27]—orienting the two lobes (Extended Data Fig. 9a,b). Ub(T) conformations differ in states 1 to 7 by 120° rotation and 35 Å translation of Ub(T) that are accompanied by conformational changes in the Ub(T) C-terminal tail between states 1 to 3 and states 4 to 7 that are illustrated by comparing states 1 and 7 (Fig. 4c,d and Extended Data Fig. 9). Conformations for Ub(T) fully transitioned to its binding site on the C-lobe were not observed[25–27], perhaps because Ub(T) remains covalently bound to E2 and E3[HECT] active site thiols or because the C-terminal tail of E3[HECT] that interacts with Ub(T) after it binds the C-lobe remains coordinated by the E3[HECT] C-lobe[25,27,35,36] (Extended Data Fig. 9b). Although the E3[HECT]–Ub(T) product is not observed, Ub(T) conformations through states 1 to 7 show Ub rotating and moving toward the C-lobe with states 5 to 7 approaching conformations observed in structures of Ub(T) bound to the C-lobe (Extended Data Fig. 9). Together, these observations suggest that our structures represent early intermediates during transthiolation before Ub(T) binds the C-lobe.

The predominant Ub conformation observed in reconstructions of E2–Ub(T)–E3[HECT] shows Ub(T) densities proximal to the crossover helix of E2 (Fig. 4c,d). This Ub(T) conformation is evident in several reconstructions (Extended Data Fig. 8d), but only state 1 shows unambiguous densities for the Ub C-terminal tail. The closed conformation observed for Ub(T) and E2 in state 1 is distinct from closed conformations reported for activated E2–Ub(T) mimics when bound to RING-type E3 ligases[21,37–40]. Further, this configuration for E2–Ub(T) differs from prior structures of HECT or RING E3s with E2-Ub(T) mimics because it shows Ub(T) C-terminal amino acids within a hydrophobic groove formed by remodelling E2 amino acid residues 86–92 to create interactions between E2, Ub(T) and E3[HECT] that are unique to this complex (Fig. 4d,f). The conformation of E2–Ub(T) in state 1 appears to be specific to complexes with E3[HECT], as this configuration was not evident for any reconstructions determined for E1–Ub(T)–E2 (Extended Data Figs. 6 and 7). Electron microscopy densities for Ub(T) are less resolved in states 2 to 7, suggesting increased flexibility as Ub(T) is released from its position in state 1. Further, states 2 to 7 show the E2 loop between residues 86–92 in its canonical conformation with fewer interactions between E2 and E3[HECT] as Ub(T) transitions toward the E3[HECT] C-lobe as exemplified in state 7 (Fig. 4d,f).

## E2–Ub(T) contacts contribute to E1 or E3 transthiolation

To assess contributions of amino acid side chains observed within interfaces of E2, E3[HECT] and Ub(T) that are unique to state 1 or differ between states (Fig. 4d–f), we introduced point substitutions into Ub, E2 or E3[HECT] and assayed the proteins in pulse–chase transthiolation assays for transfer of fluorescent Ub(T) from E2-Ub(T) to E3[HECT] (Fig. 4g, Extended Data Fig. 9c and Supplementary Fig. 4).

Interactions between Ub(T) and E2 that are unique to state 1 and absent from state 7 include contacts between Ub(T) and the E2 crossover helix and contacts formed by accommodating the Ub(T) C-terminal tail into a hydrophobic groove in E2 involving Ub(T) Val70, Leu71, Leu73 and Arg74 (Fig. 4f). Consistent with contacts in state 1 between the E2 crossover helix and Ub(T), individual substitution of E2 residues Leu97, Lys101, Ser105, Ser108 and Leu119 to remove the side chains or replace them with bulkier side chains each decreased transthiolation to E3[HECT] (Fig. 4g). With respect to contacts to the Ub(T) C-terminal tail, alanine substitution of Ub(T) Val70, Leu71 or Leu73 decreased activity (Fig. 4g). Whereas Ub(T) Leu71 and Leu73 contact E3 after Ub(T) transfer to the C-lobe, Val70 does not[25]. In state 1, the Ub(T) Arg74 side chain guanidinium interacts with Asp117 of E2, the carbonyl oxygen of E3[HECT] Ala636, and the cyanomethyl group of our linker, a surrogate for the tetrahedral intermediate oxyanion of Ub(T) Gly76 (Fig. 4e,f). Consistent with contacts in state 1, Ub(T) R74A diminished transthiolation to E3[HECT] (Fig. 4g). In addition, backbone nitrogen and carbonyl oxygen of Ub(T) Gly75 are within hydrogen-bond distance to the backbone carbonyl oxygen and nitrogen of active site Cys85 of E2, and these interactions are only present in state 1. These data are concordant with prior results indicating that residues in the Ub C-terminal tail contribute to E2-to-E3 transthiolation[25,41,42].

Interactions between E3[HECT] and E2-Ub(T) are extensive in state 1, and include contacts with conserved E3[HECT] residues such as Asp617, His637, Thr638, Phe640, Arg642 and Val627 within an interface formed by E2 residues Asn81, Ser83 and Leu119 (Fig. 4f). E3[HECT] Thr638 is proximal to Ub(T) Arg74 in state 1 and Gly75 in state 7 and a T638A reduced transthiolation (Fig. 4g). Val627, Arg642 and His637 form an E3 surface that is juxtaposed with an E2 loop between residues 117–121 in both states and individual Ala substitution of these E3[HECT] residues and of E2 Leu119 resulted in diminished transthiolation activity (Fig. 4g). Side-chain contacts between Asp617 and E2 Asn81 and Ser83 are closer in state 1 relative to state 7 and E3[HECT] D617A substitution diminished activity, as did E2 N81A and S83A substitutions (Fig. 4g). Phe640 is highly conserved among E3[HECT] ligases and was previously identified as important for activity, however its role remained somewhat unclear because it is 7 Å from the Ub(T) Gly76 and E2 active site in prior structures[25]. In state 1, the Phe640 side chain is within 4 Å of Ub(T) Gly76 and the E3[HECT] thiol (Cys639) opposite the E2 thiol (Cys85) adjacent to E2 Leu119, Asn79 and Asn81, where it buttresses the active site to support side-chain configurations only observed in state 1 (Fig. 4f). Consistent with these contacts, F640A substitution diminished transthiolation activity by 30-fold (Fig. 4g).

## Roles of E2 surfaces in transthiolation

Discrete closed conformations of E2–Ub(T) where Ub(T) contacts the E2 crossover helix are observed in both E1 and E3 complexes. For E1, this E2–Ub(T) conformation with Ub(T) in the acceptor position is associated with the product of transthiolation (Fig. 2). For E3, this E2–Ub(T) conformation with Ub(T) in the donor position is associated with the substrate of transthiolation (Fig. 4). Mutations that stabilize closed conformations of E2–Ub(T) might enhance E1-to-E2 transthiolation but inhibit E2-to-E3[HECT] transthiolation. Destabilizing mutations could have the reverse effect. To test this, we designed E2 mutations and assayed proteins for transthiolation with E1 or E3[HECT] based on contacts in the Ub(T) acceptor conformation (cluster 5, state 10) of E1–Ub(T)–E2/Ub(A)-AMP and Ub(T) donor conformation (state 1) of E2–Ub(T)–E3[HECT], referred to hereafter as the E1–E2 Ub(T) acceptor complex and E2–E3[HECT] Ub(T) donor complex, respectively (Fig. 5a,b and Supplementary Figs. 4 and 5).

E2 residues Leu119, Ser83 and Asn81 establish unique contacts in complexes with E1 and E3. Leu119 is adjacent to E1 and E3 active sites and L119A substitution decreased transthiolation with E1 and E3[HECT] (Fig. 5c,d). By contrast, E2 Ser83 and Asn81 contact E3[HECT] but not E1. Consistent with this, S83A and N81A decreased E2–E3[HECT] transthiolation

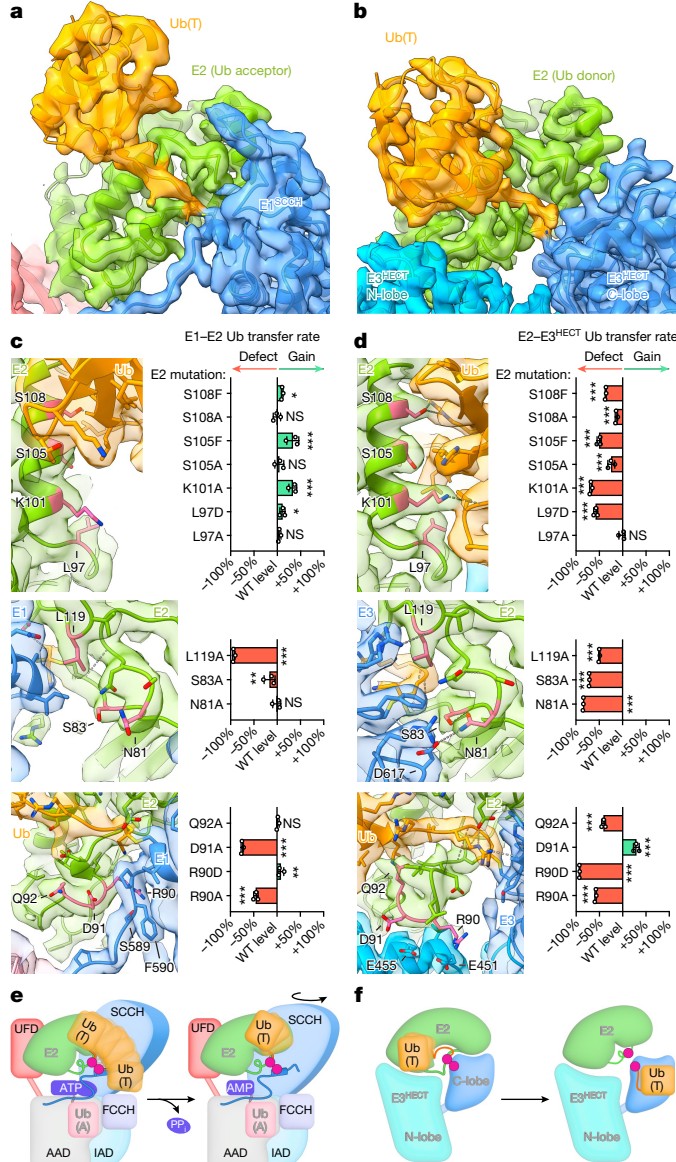

**a,b**

Ub(T)

E2 (Ub acceptor)

E1^SCCH

Ub(T)

E2 (Ub donor)

E3^HECT N-lobe

E3^HECT C-lobe

**c** E1–E2 Ub transfer rate

Defect ← → Gain

E2 mutation:
S108F *
S108A NS
S105F ***
S105A NS
K101A ***
L97D *
L97A NS

−100% −50% WT level +50% +100%

E2 Ub
S108
S105
K101
L97

L119A ***
S83A ***
N81A NS

−100% −50% WT level +50% +100%

E1 E2
L119
S83
N81

Q92A NS
D91A ***
R90D **
R90A ***

−100% −50% WT level +50% +100%

Ub E1
Q92
D91
R90
S589
F590

**d** E2–E3^HECT Ub transfer rate

Defect ← → Gain

E2 mutation:
S108F ***
S108A ***
S105F ***
S105A ***
K101A ***
L97D ***
L97A NS

−100% −50% WT level +50% +100%

E3 E2
S108
S105
K101
L97

L119A ***
S83A ***
N81A ***

−100% −50% WT level +50% +100%

E3 E2
L119
S83
N81
D817

Q92A ***
D91A ***
R90D ***
R90A **

−100% −50% WT level +50% +100%

Ub E2
Q92
D91
R90
E3
E455 E451

**e** UFD, Ub(T), SCCH, E2, Ub(T), ATP, Ub(A), FCCH, AAD, IAD → AMP, PP_i

**f** E2, Ub(T), C-lobe, E3^HECT, N-lobe → E2, Ub(T), E3^HECT, N-lobe

**Fig. 5 | Differential contribution of E2 residues to transthiolation when E2 is the acceptor of Ub(T) from E1 or donor of Ub(T) to E3^HECT.** **a,b**, Reconstructions and models for E1–Ub(T)–E2 with Ub(T) in the acceptor position (state 10) (**a**) and for E2–Ub(T)–E3^HECT with Ub(T) in the donor position (state 1) (**b**), highlighting distinct contacts of E2 with Ub(T) and E1 or E3. **c**, Electron microscopy densities and models and results of E1-to-E2 transthiolation assays with indicated E2 mutants. Bar graphs depict loss or gain of activity relative to wild type ($n = 18$ replicates) and represent mean ± s.d of $n = 3$ replicates. Statistical differences between wild type and mutants by two-tailed unpaired $t$-test, ***$P < 0.001$, **$P < 0.01$, *$P < 0.05$; NS, not significant. **d**, As in **c** but for E2-to-E3 transthiolation assays with indicated E2 mutants. Bar graphs depict loss or gain of activity relative to wild type ($n = 9$ replicates) and represent mean ± s.d. of $n = 3$ replicates (Supplementary Figs. 4 and 5). Models and electron microscopy densities colour-coded according to Figs. 2 and 4, with mutated residues in pink. Isosurface levels contoured at 0.57 and 0.51 (**a,c**) and 0.71 (**b,d**). **e**, Schematic for E1-to-E2 transthiolation indicating Ub(T) conformations during transthiolation and adenylation. **f**, Schematic for E2-to-E3 transthiolation indicating Ub(T) conformations and E2 loop remodelling during transthiolation. Domains are coloured and labelled, active site Cys in E1, E2 and E3 are indicated as magenta circles.

a negative effect on E2–E3^HECT transthiolation but elicited little effect on E1–E2 transthiolation (Fig. 5c,d). Ser105 and Ser108 are in the E2 crossover helix near Ub(T) in the E2–E3^HECT Ub(T) donor complex but do not contact Ub(T) in the E1–E2 Ub(T) acceptor complex. S105F and S108F were predicted to stabilize interactions with Ub(T) via contacts with hydrophobic residues on Ub(T) surface, and each stimulated transthiolation from E1 to E2 and inhibited transthiolation from E2 to E3 (Fig. 5c,d).

The E2 loop that is remodelled in complex with Ub(T)–E3 includes Arg90, Asp91 and Gln92, a region of E2 that is proximal to the E1 crossover loop in E1–Ub(T) (Fig. 5c,d). E2 Gln92 contacts Ub(T) in the E2–E3^HECT donor complex but not in the E1–E2 acceptor complex, and Q92A substitution exerted a negative effect on E2–E3^HECT transthiolation relative to E1–E2 transthiolation (Fig. 5c,d). Densities for the E2 Arg90 side chain are weak in the E2–E3^HECT donor complex, but its side chain projects towards Glu455 and Glu451 of E3^HECT (Fig. 5d). Consistent with this, charge reversal by E2 R90D substitution diminished E2–E3^HECT transthiolation, an effect that was partially rescued by introducing E455Q and E451Q charge-neutralizing mutations in E3^HECT (Extended Data Fig. 9d and Supplementary Fig. 4f). E2 Arg90 and Asp91 contact E1 Ser589 and Phe590, residues adjacent to the E1 catalytic cysteine in the E1–E2 Ub(T) acceptor complex and R90A or D91A substitutions decreased E1–E2 transthiolation (Fig. 5c). Asp91 is proximal to E3^HECT Glu455 and Glu451 in state 1 in the E2–E3^HECT donor complex, and D91A stimulated E2–E3^HECT transthiolation, perhaps because substitution removes unfavourable electrostatic interactions in this interface (Fig. 5d). Collectively, these data underscore unique elements within E2 that contribute differentially to E1–E2 or E2–E3^HECT transthiolation.

## Discussion

Isoenergetic transthiolation reactions are easily reversed, so the Ub-conjugation system employs different strategies to promote forward reactions. For E1–E2 transthiolation, the reaction is driven forward by coupling adenylation of Ub(A) with E1 domain movements that facilitate interactions between Ub(T) and E2 during transfer. That Ub(A) adenylation and PP_i•Mg^{2+} release can occur during transthiolation suggests that once E2-Ub(T) is formed, the E1 and Ub(A)-adenylate are primed to form the next E1-Ub(T) thioester bond, which drives the reaction forward and prevents E2-to-E1 transthiolation in reverse (Fig. 5e). Coupling of adenylation and transthiolation is also consistent with early observations that consumption of E2-Ub(T) by E1 in the presence of excess AMP and PP_i can generate ATP and E1-Ub(T)[18,19]. Further, recent studies identified Ub E1 mutations associated with human diseases, including VEXAS (vacuoles, E1 enzyme, X-linked, autoinflammatory, somatic) syndrome[43–45], that include mutations that retain adenylation activity but are defective in E1–E2 transthiolation[45]. Of note, and consistent with coupling of adenylation and transthiolation, these mutations are located near the adenylation active site, more than 30 Å from the transthiolation site, at interfaces that alter configuration during transthiolation (Extended Data Fig. 10a–d).

Unlike E1-to-E2 transthiolation, which is driven forward by coupling adenylation and transthiolation, E2-to-E3^HECT transthiolation promotes directionality by remodelling residues around the E2 active site that depend on unique conformations of the C-terminal Ub(T) tail, its linkage to the E2 and E3 catalytic cysteines, and surfaces of E3^HECT (Fig. 5f). After transfer is initiated, E2 residues adopt canonical conformations to release the Ub(T) C-terminal tail, enabling capture of Ub(T) by the E3^HECT C-lobe[25,27]. The importance of Ub(T) and its thioester linkage to E2 is consistent with prior studies of the E3^HECT ligase E6AP which measured a 100-fold preference for binding of E2-Ub(T) over uncharged E2[46]. Further, the disulfide-linked E2-Ub76C thioester mimic bound E6AP with affinities similar to uncharged E2[47,48], suggesting that E2-Ub(T) recognition requires a native thioester-linked Ub(T). Our structures support a hypothesis that E3^HECT discriminates E2-Ub(T) from uncharged E2 indirectly through recognition of the unique interface between E2,

but elicited no defect in E1–E2 transfer. E2 Leu97 and Lys101 contact Ub(T) in the E2–E3^HECT Ub(T) donor complex but make no contacts in the E1–E2 Ub(T) acceptor complex. Substituting these residues had

Ub(T) and E3^HECT that is created by accommodating the Ub(T) C-terminal tail within remodelled E2 loop residues between amino acids 86–92 as observed in state 1. As this conformation is dependent on the Ub(T) tail and covalent linkage of Ub(T) Gly76 to E2, remodelling of E2 residues might prohibit the reverse reaction after release of the Ub(T) tail as Ub(T) transitions toward the acceptor interface on the E3^HECT C-lobe. The conformation observed for E2–Ub(T)–E3^HECT in state 1 may also be relevant to pathogenic bacterial HECT-type E3 transthiolation ligases[49,50] as studies on SspH2 and SspH1 reveal selective binding to E2-Ub(T) and NMR chemical shift perturbations between E2 and Ub in E2-Ub(T) within surfaces that overlap with contacts observed in our closed configuration for E2–Ub(T)–E3^HECT. Further, contacts observed in state 1 of E2–Ub(T)–E3^HECT involving the E2 loop and residues between E2 and E3 are highly conserved in human E2s that form functional pairs with HECT-type E3 ligases[51] (Extended Data Fig. 10e).

The chemical biology approach described here enabled isolation and visualization of more stable mimics of otherwise transient transthiolation complexes between native Ub enzymes including E1, E2 and E3^HECT proteins. New intermediates were revealed in these processes, but our structures preclude full atomic descriptions for states prior to or after transfer; thus further work will be required to capture a full continuum of substrates, transition states, and products during transthiolation. The strategy outlined here for capturing transthiolation mimics for E1 and E3^HECT provides a structural basis for mutations in human disease, and it should prove useful in resolving intermediates for other Ub or Ubl enzymes and in the development of chemical tools to inhibit or co-opt the Ub system for therapeutic benefit.

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

## Methods

### Cloning

Cloning of *S. pombe* Uba1, Cys-Ub with an N-terminal Ulp1-cleavable His$_6$-Smt3 tag, and Ub without tags was described previously[15,17,52]. All mutations were generated using PCR-based mutagenesis. Constructs encoding full-length *S. pombe* Ubc4(C21S/C107S) with a C-terminal Gly–Gly linker followed by thrombin-cleavable His$_6$ tag, and constructs for full-length *S. pombe* Ubc4 with C-terminal His$_6$ tag were constructed using the pET29b vector (Novagen). The DNA encoding *S. pombe* Pub2$^{HECT}$ (amino acid residues 295–671) was inserted into the pSMT3 vector to introduce an N-terminal Ulp1-cleavable[53] His$_6$ Smt3 tag. The DNA encoding *S. pombe* Ub lacking the last two amino acids at its C terminus, with StrepTag and TEV protease cleavage site (MWSHPQF EKSAENLYFQGSGG) added at its N-terminus, further referred to as Ub(–2), was inserted into vector pTXB1 (New England Biolabs), to generate a C-terminal fusion with an Mxe intein-chitin binding domain.

### Protein expression and purification

DNA plasmids encoding recombinant proteins were expressed in *Escherichia coli* strain BL21 (DE3) codon plus (Stratagene). To produce Pub2$^{HECT}$, cells were grown in Superbroth at 37 °C to $OD_{600}$ = 0.8, cooled in an ice/water bath for 20 min before addition of 1-thio-β-D-galactopyranoside (IPTG) to a final concentration of 0.4 mM. Cultures were incubated for 14–18 h at 18 °C. Uba1, Ubc4, Cys-Ub and untagged Ub were expressed as described previously[15,17,52]. Cells were collected by centrifugation at 4,000*g* (Beckman JLA-8.1000) for 20 min at 4 °C. Cell pellets were resuspended in 50 mM sodium HEPES pH 8.0, 350 mM NaCl, 20% sucrose and snap frozen in liquid nitrogen. Cell pellets were thawed, supplemented in lysis buffer containing 0.5 mM TCEP, 2.5 mM MgCl$_2$, 0.1 mg ml$^{-1}$ DNAse I, 1 mg ml$^{-1}$ lysozyme, 1 mM PMSF, and lysed by sonication (2 s on, 8 s off, 50% output amplitude) twice for 3 min with a 3 min break in between using Digital Sonifier 450 Cell Disruptor (Branson) on ice/water bath. Lysates were clarified by centrifugation at 47,000*g* (Beckman JA-20).

For purification of His$_6$-tagged proteins, supernatant lysates were mixed with 15 mM imidazole and applied to Ni$^{2+}$-NTA superflow resin (Qiagen) by gravity flow. Beads were washed with buffer containing 20 mM sodium HEPES pH 8.0, 350 mM NaCl, 0.2 mM TCEP, and 20 mM imidazole at 4 °C. Proteins were eluted in buffer containing 20 mM sodium HEPES pH 8.0, 350 mM NaCl, 0.2 mM TCEP, and 250 mM imidazole at 4 °C. His$_6$–Smt3 tags used for affinity purification were removed by incubation with Ulp1[53] and separated by size-exclusion chromatography (SEC) with Pub2 and Ubc4 separated using HiLoad 26/600 Superdex 75 PG column and Uba1 separated using HiLoad 26/600 Superdex 200 PG column (GE Healthcare), equilibrated in 20 mM sodium HEPES pH 7.5, 250 mM NaCl, 0.1 mM TCEP at 4 °C. For Ubc4(C21S/C107S), the C-terminal His$_6$ tag was removed by treatment with thrombin then separated by SEC. After SEC, Uba1 was further purified by anion-exchange chromatography using a MonoQ 10/100 GL column (GE Healthcare) in 20 mM sodium HEPES pH 8.0, 0.1 mM TCEP and a linear gradient of 50–800 mM NaCl at 4 °C. To obtain fluorescein-Ub variants, Cys-Ub variants containing an additional N-terminal Cys for labelling were obtained as described previously[15]. Purified Cys-Ub variants were incubated in 20 mM HEPES pH 7.5, 200 mM NaCl, 2 mM TCEP for 10 min at room temperature, desalted into the same buffer without TCEP and modified by adding 10-fold molar excess of 5-fluorescein maleimide (Thermo Fisher Scientific) and incubating for 2 h at room temperature. Reactions were then buffer exchanged to 20 mM sodium HEPES pH 7.5, 200 mM NaCl using 7 kDa MWCO Zeba Spin Column (Thermo Fisher Scientific) and separated by size exclusion using Superdex 75 Increase 10/300 GL column (GE Healthcare) equilibrated in the same buffer at 4 °C. Native Ub (without affinity tags), was purified as described previously[52], with additional SEC using a HiLoad 26/600 Superdex 75 prep grade column (GE Healthcare) equilibrated in 20 mM Tris•HCl pH 7.5, 250 mM NaCl at 4 °C. Fractions containing the desired proteins were pooled, concentrated, flash-frozen in liquid nitrogen, and stored at −80 °C.

### Preparation of Ub-PSAN

H$_2$N-Gly–PSAN was synthesized as described[28]. Ub$^{1-74}$ fused to Mxe intein-chitin binding domain was expressed, and cells collected and lysed using the same procedure as for Pub2$^{HECT}$. Lysate supernatant was incubated with chitin beads (New England BioLabs) equilibrated in 30 mM sodium HEPES pH 8.0, 350 mM NaCl. Beads were washed with ten bed volumes of equilibration buffer and then two bed volumes of 30 mM Bis-Tris•HCl pH 6.5, 350 mM NaCl at 4 °C. The Ub$^{1-74}$-2-mercaptoethanesulfonate (MESNa) thioester was obtained by incubating the resin in 2 bed volumes of 30 mM Bis-Tris•HCl pH 6.5, 350 mM NaCl, 200 mM MESNa at room temperature for 12–16 h. The cleaved protein was eluted and transferred to ice/water bath, then the resin was treated again with cleavage buffer for 12 h at 4 °C, followed by elution. Eluted fractions were combined, concentrated using 3 kDa MWCO Amicon filter (Milipore) to 8–10 mg ml$^{-1}$ and treated with 1 M hydrazine at 30 °C for 30 min. The resulting Ub$^{1-74}$ hydrazide was separated by SEC using HiLoad 26/600 Superdex 75 prep grade column (GE Healthcare) equilibrated with 25 mM sodium phosphate pH 6.5, 350 mM NaCl at 4 °C. Fractions containing Ub$^{1-74}$ hydrazide were pooled, concentrated, flash-frozen in liquid nitrogen, and stored at −80 °C.

To generate Ub$^{1-74}$ acyl azide for coupling with H$_2$N-Gly–PSAN, Ub$^{1-74}$ hydrazide, at a concentration of 3 mM, was combined with 100 mM sodium citrate pH 3.0 on ice/water bath. NaNO$_2$ was added to a final concentration of 50 mM (from 0.5 M stock adjusted to pH 5 with HCl on ice/water bath immediately before use) and the reaction transferred to −20 °C (ice:NaCl bath at 3:1 ratio) for 5 min. An equal volume of 250 mM H$_2$N-Gly–PSAN (HCl salt) in 1 M sodium HEPES pH 8 was added immediately after dissolving and the reaction incubated for 10 min on ice/water bath followed by 10 min at room temperature, and buffer exchanged into 20 mM sodium HEPES pH 8, 350 mM NaCl at 4 °C using a 7 kDa MWCO Zeba Spin Column (Thermo Fisher Scientific). Ub-PSAN was separated by SEC using Superdex 75 Increase 10/300 GL column (GE Healthcare) equilibrated in 20 mM sodium HEPES pH 8, 350 mM NaCl at 4 °C. The molecular weight of Ub-PSAN was confirmed by ultraperformance liquid chromatography electrospray mass spectrometry using Acquity UPLC-MS (Waters). Ub-PSAN was concentrated to ~1 mM, flash-frozen in liquid nitrogen, and stored at −80 °C.

### Preparation of E1–Ub(T)–E2 and E2–Ub(T)–E3$^{HECT}$ complexes

To obtain E2–Ub vinyl thioether for reactions with E1 and E3$^{HECT}$, a mixture of 0.6 mM Ub-PSAN, 0.4 mM Ubc4(C21S/C107S), 20 mM sodium HEPES pH 8.0, 200 mM NaCl was incubated for 1 h at 4 °C. The E2–Ub vinyl thioether product was separated by SEC on a HiLoad 26/600 Superdex 75 prep grade column (GE Healthcare) equilibrated in the same buffer.

To obtain E1–Ub(T)–E2, a mixture of 75 μM E2–Ub vinyl thioether, 50 μM Uba1 (lacking the first 12 amino acids), 20 mM sodium HEPES pH 8.0, 100 mM NaCl, 0.2 mM TCEP was incubated for 1 h at room temperature, and then incubated on ice/water bath. The 30 ml total reaction volume was divided into four 7.5 ml portions. Each portion was applied to a 5 ml StrepTrap HP column (GE Healthcare) equilibrated in Tris•HCl pH 7.2, 200 mM NaCl, 0.2 mM TCEP at 4 °C. After washing with 10 column volumes of equilibration buffer, proteins were eluted in the same buffer containing desthiobiotin at a final concentration of 2.5 mM. Eluted fractions from all runs were combined and incubated with TEV protease for 9 h at 4 °C to remove the StrepTag–TEV tag from Ub. After tag cleavage, E1–Ub(T)–E2 was separated by anion-exchange chromatography using MonoQ 5/50 GL column (GE Healthcare) in 20 mM Tris•HCl pH 7.2, 0.1 mM TCEP and a linear gradient of 50–400 mM NaCl at 4 °C. Fractions containing E1–Ub(T)–E2 were concentrated to ~12 mg ml$^{-1}$, flash-frozen in liquid nitrogen, and stored at −80 °C.

To obtain E2–Ub(T)–E3$^{HECT}$, a 30 ml solution containing 75 µM E2–Ub vinyl thioether, 75 µM Pub2$^{HECT}$, 20 mM sodium HEPES pH 8.0, 200 mM NaCl, 0.5 mM TCEP was incubated for 8 h at room temperature, then moved to ice/water bath. The reaction mixture was subjected to StrepTag affinity chromatography and TEV protease cleavage as described above for E1–Ub(T)–E2 complex. After cleavage, E2–Ub(T)-E3$^{HECT}$ was isolated with two successive rounds of anion-exchange chromatography using a MonoQ 5/50 GL column (GE Healthcare) equilibrated with 20 mM Tris•HCl pH 7.5, 0.1 mM TCEP with a linear gradient of 50–280 mM NaCl at 4 °C. Fractions containing E2–Ub(T)–E3$^{HECT}$ were concentrated to ~13.5 mg ml$^{-1}$, flash-frozen, and stored at −80 °C.

## Cryo-EM sample preparation and data collection

Prior to grid preparation, an aliquot of E1–Ub(T)–E2 or E2–Ub(T)–E3$^{HECT}$ was rapidly thawed in a room temperature water bath and centrifuged for 10 min at 4 °C at 18,000g. E1–Ub(T)-E2 was diluted to 3 mg ml$^{-1}$ in Tris•HCl pH 7.2, 100 mM NaCl and preincubated for 30 min on ice/water bath with 5 mM MgCl$_2$, 1 mM ATP, and ~0.8 molar equivalents of Ub. E2–Ub(T)–E3$^{HECT}$ was diluted to 4.5 mg ml$^{-1}$ in Tris•HCl pH 7.2, 100 mM NaCl. Prior to vitrification, CHAPSO was added to samples at final concentrations of 0.05% and 0.1% w/v for E1–Ub(T)–E2 and E2–Ub(T)–E3$^{HECT}$ complexes, respectively. Four microlitres of sample was applied to freshly glow-discharged UltrAuFoil 300 mesh R1.2/1.3 grids (Quantifoil) at 100% humidity at 25 °C. After 8 s, samples were blotted for 3.5–4.0 s and vitrified by plunging into liquid ethane using a Vitrobot Mark IV (FEI-Thermo Fisher).

Cryo-EM data were collected at MSK Richard Rifkind Center for Cryo-EM, using a Titan Krios transmission electron microscope (FEI-Thermo Fisher) operated at 300 keV. Cryo-EM movies (40 frames per movie, 4 s exposure time) were recorded at a dose rate of ~20 e$^-$ pixel$^{-1}$ s$^{-1}$ using a K3 Summit direct electron detector (Gatan) operated in super-resolution mode at a physical pixel size of 1.064 Å. Automated data collection was performed in Serial EM[54] using image shift to record data from nine ice holes per stage movement. Two datasets for E1–Ub(T)–E2 were obtained with 12,132 and 14,545 movies each, and 2 datasets for E2–Ub(T)–E3$^{HECT}$ were obtained with 15,705 and 14,101 movies each.

## Cryo-EM image processing

Initial data processing steps were similar for E1–Ub(T)–E2 and E2–Ub(T)–E3$^{HECT}$. Movie frames from each session were gain normalized, 2× Fourier cropped, aligned and summed with and without dose-weighting using MotionCor2[55]. Estimation of the contrast transfer function (CTF) was performed using Gctf[56] from non-dose weighted micrographs. Micrographs with estimated resolution limits worse than 4.5 Å, poor CTF fit or with crystalline ice were discarded. Initial particle sets were obtained by reference-free auto-picking with Laplacian-of-Gaussian filtering in RELION 3.1[57,58]. Subsequent steps were performed in cryoSPARC 4.0.2[59] with the exception of particle picking using Topaz[60] and Bayesian polishing performed in RELION 3.1[61]. UCSF Pyem was used to convert particle metadata from cryoSPARC to RELION format[62]. Particles from each dataset underwent several rounds of 2D classification to remove junk particles and to obtain subsets from 2,000 random micrographs for training neural network-based particle picking in Topaz. Trained models were applied to full datasets, and identified particles were extracted using a 256-pixel box size.

For E1–Ub(T)–E2, Topaz-picked particles were used in rounds of 2D classifications to remove junk particles. Two successive rounds of ab initio 3D reconstruction and heterogeneous refinement were performed, first with three classes, then with four, each time removing classes lacking secondary structure features. 1,826,497 particles were re-extracted using a 384-pixel box size and pooled for a single ab initio 3D reconstruction followed by non-uniform 3D refinement[63], resulting in a consensus map with a nominal resolution of 3.0 Å. To remove low-quality particles, four consecutive rounds of heterogeneous

refinement were initialized with four copies of a consensus map low-pass filtered to 15 Å and one copy lowpass filtered to 30 Å. Particles assigned to the lowest resolution class were discarded after each round. Particles were recentred and subjected to non-uniform 3D refinement and Bayesian polishing, followed by 2D classification to remove images with artefacts resulting in 1,610,345 particles that yielded a consensus map with a nominal resolution of 2.64 Å, which improved to 2.51 Å after per-particle defocus refinement, per optics group estimation of the beam tilt, trefoil, spherical and fourth-order aberrations, followed by second round of per-particle defocus refinement. Particles were next subjected to heterogeneous refinement (with 6 classes and 2× binning) initialized with 20 Å lowpass-filtered consensus map resulting in four classes with Ub(A) and two without Ub(A) that were combined to yield two particle sets for doubly and singly loaded complexes (1,295,595 and 314,750 particles respectively), which after non-uniform 3D refinement, resulted in consensus maps at nominal resolutions of 2.50 and 2.79 Å, respectively. To resolve Ub(T) conformations in doubly and singly loaded complexes, particles were then subjected to 3D classification without image realignment (10 classes, target resolution 7 Å, number of O-EM epochs 8, O-EM learning rate init 0.3, initialization mode principal components analysis (PCA), force hard classification on) with a mask focused on Ub(T) region. Maps were generated by 3D reconstruction using particles from each class and their alignment information and a half-set splits from the gold-standard refinement of their parental sets (consensus maps). The 3D classes used for model building were further subjected to non-uniform 3D refinement. The continuum of E1 SCCH rotation relative to E1 adenylation domains (IAD and AAD) was resolved by 3D variability analyses[64] (3 modes to solve, filter resolution 5 Å) performed separately for particles corresponding to doubly and singly loaded complexes, using a mask encompassing the E1–E2 region but excluding Ub(T) regions. In both cases, particles were sorted into five clusters based on values of the latent coordinate of the component 1, best capturing rotation of E1 SCCH domain. Five clusters were chosen because this number adequately tracked SCCH movement (1.5° to 2° of rotation per cluster) while ensuring sufficient particles within each cluster for further analysis. Particles from each cluster were subjected to non-uniform refinement, resulting in maps with nominal resolutions of 2.95–3.16 Å for singly loaded complexes and 2.67–2.86 Å for doubly loaded complexes. Ub(T) conformations in each cluster were resolved through 3D classifications without image realignment (10 classes, target resolution 7 Å, number of O-EM epochs 8, O-EM learning rate init 0.3, initialization mode PCA, force hard classification on) using a focus mask on the Ub(T) region. Maps were generated by 3D reconstruction using particles from each class and their alignment information and half-set splits from the gold-standard refinement of parental sets (clusters 1–5). The 3D classes used for model building were further subjected to non-uniform 3D refinement. Where applicable, maps were lowpass filtered to 5 Å using a 10th-order Butterworth filter. Statistics for data collection are listed in Supplementary Table 1.

For E2–Ub(T)–E3$^{HECT}$, Topaz-picked particles were selected after 2D classification from each dataset and re-extracted using 320-pixel box size, combined, and subjected to rounds of 2D classification, resulting in 2,704,150 particles. Ab initio 3D reconstructions using six classes were performed, followed by heterogeneous refinement (with 2× binning). Particles from classes with defined secondary structure features (2,511,724 particles) were combined and used in single ab initio 3D reconstruction followed by non-uniform 3D refinement to a map with a nominal resolution of 3.30 Å. Particles were recentered, subjected to non-uniform 3D refinement and Bayesian polishing, followed by 2D classification to remove images with artefacts, and per-particle defocus refinement, resulting in 2,428,313 particles that yielded a consensus map with a nominal resolution of 3.09 Å. Subsequently, 3D variability analysis (3 modes to solve, filter resolution 5 Å) was performed with a mask encompassing the entire complex. Particles were then split into 20 clusters based on all solved modes of 3D variability, an empirically

determined number that resulted in best separation between complexes. Each cluster was subjected to non-uniform 3D refinement, resulting in maps with nominal resolutions of 3.27–3.56 Å. The 8 clusters resulted in maps with no apparent or poorly resolved Ub(T) density. Ub(T) densities were resolved in maps for 12 clusters with 6 maps capturing Ub(T) at positions proximal to E2 and 6 other maps with Ub(T) in positions more distal from E2 (states 2–7). All 12 maps revealed Ub(T) C-terminal residues between E2 and E3[HECT]. Particles from 6 clusters containing Ub(T) at a position most proximal to E2 (790,208 particles) were combined and used for non-uniform 3D refinement resulting in a map with a nominal resolution of 3.17 Å. To select the best particles and to improve the map at the transthiolation active site, a 3D classification without image alignment (4 classes, target resolution 7 Å, number of O-EM epochs 8, O-EM learning rate init 0.3, initialization mode PCA, force hard classification on) was performed with a mask focused on the transthiolation site. One of four classes showed improved density corresponding to the transthiolation site, including E2 amino acids around its active site. This class containing 204,763 particles was subjected to non-uniform 3D refinement to yield a map with a nominal resolution of 3.23 Å (state 1). Reported resolutions were determined using the gold-standard 0.143 criterion based on Fourier shell correlation. Statistics for data collection are listed in Supplementary Table 1.

## Model building and refinement

Initial coordinates were generated by docking individual chains from reference structures into cryo-EM maps in UCSF Chimera[65] followed by manual building in Coot[66]. For E1–Ub(T)–E2, the crystal structures of Uba1–Ubc4/Ub/ATP·Mg and Ub (Protein Data Bank (PDB): 4II2[17] and 6O82[15], respectively) were used. For E2–Ub(T)–E3[HECT], a homology model of Pub2[HECT] was obtained from SWISS-MODEL[67] based on crystal structures of UbcH5B~Ub-NEDD4L (PDB: 3JW0[25]), Ubc4 and Ub (PDB: 4II2[17] and 6O82[15], respectively). Coordinates for all models were produced via iterative rounds of refinement and building in real space using Phenix and Coot[66,68]. Geometry restraints for the linker (transthiolation intermediate analogue) were generated using Phenix.elbow[69]. MolProbity was used to evaluate model integrity[70]. Structure representations were generated using UCSF ChimeraX[71]. 2D slice views of electron microscopy maps were visualized using IMOD 4.11[72]. Statistics for model refinement are listed in Supplementary Table 1.

## Preparation of singly loaded fluorescein-Ub~E1 complex

To generate thioester-linked fluorescein-Ub~E1 complex, 125 µl reaction containing full-length Uba1 (8 µM), substochiometric amount of fluorescein-Ub (6 µM) in 20 mM sodium HEPES pH 7.5, 100 mM NaCl, 0.1 mM TCEP, 5 mM MgCl$_2$ and 1 mM ATP was incubated at 10 min at 30 °C. The mixture was then cooled to 4 °C and separated by anion-exchange chromatography using MonoQ 5/50 GL column (GE Healthcare) in 20 sodium HEPES pH 7.5, 0.1 mM TCEP and a linear gradient of 100–400 mM NaCl. SDS–PAGE analysis of purified fluorescein-Ub~E1 incubated with and without β-mercaptoethanol confirmed its sensitivity to reducing agent, consistent with a singly loaded fluorescein-Ub~E1 and without Ub bound non-covalently to the adenylation site.

## E1–E2 single-turnover Ub thioester transfer assays

The Ub-AMP mimic (Ub-AMSN) was synthesized as described previously[14,15,34]. Single turnover assays were performed on ice/water bath. For each replicate, singly loaded fluorescein-Ub~E1 complex (10 nM) was freshly prepared and preincubated in 20 mM sodium HEPES pH 7.5, 100 mM NaCl, 0.1 mM TCEP, 0.1 mg ml⁻¹ ovalbumin, either alone or with addition of ATP (4 mM), Ub and ATP (10 µM and 4 mM, respectively), Ub-AMSN (10 µM), or Ub-AMSN and PP$_i$ (10 µM and 4 mM, respectively) in 400 µl for 5 min. A control without addition of Ubc4 (E2) was obtained by diluting a 100 µl aliquot with an equal volume of 20 mM sodium HEPES pH 7.5, 100 mM NaCl, 5 mM MgCl$_2$, 0.1 mM TCEP, 0.1 mg ml⁻¹

ovalbumin. Thioester transfer (chase) was initiated by diluting a 100 µl aliquot with an equal volume of 100 nM E2, 20 mM sodium HEPES pH 7.5, 100 mM NaCl, 5 mM MgCl$_2$, 0.1 mM TCEP, 0.1 mg ml⁻¹ ovalbumin. The resulting concentrations at time 0 were 5 nM for fluorescein-Ub~E1 complex and 50 nM for E2. Aliquots from indicated timepoints and a control without E2 were quenched by addition of LDS NuPAGE buffer supplemented with EDTA (final concentrations of 1× and 50 mM, respectively). Products were separated on 4–12% NuPAGE BIS-Tris gels with 1× MOPS running buffer (Thermo Fisher Scientific). To increase fluorescence signal by converting fluorescein to its di-anionic form, gels were incubated in 50 mM Tris•HCl pH 9.5 for 3 min prior to scanning using Amersham Typhoon 5 (Cytiva). Bands were quantified using ImageQuant software (Cytiva).

## E1–E2 multiple turnover Ub thioester transfer assays

E1–E2 Ub thioester transfer assays were performed in 160 µl reactions with 1.5 nM full-length Uba1, 200 nM of indicated variant of Ubc4, 5 µM Ub, 20 mM sodium HEPES pH 7.5, 50 mM NaCl, 5 mM MgCl$_2$, 0.1 mM TCEP. Reactions were conducted at 22 °C and initiated by adding ATP to a final concentration of 200 µM including a control that lacked ATP. Aliquots were removed at indicated timepoints and quenched by addition of LDS NuPAGE buffer supplemented with EDTA (final concentrations of 1× and 50 mM, respectively) and products were separated on 12% NuPAGE BIS-Tris gels with MOPS running buffer (Thermo Fisher Scientific). Gels were stained using Flamingo dye (Bio-Rad), scanned using Amersham Typhoon FLA 9500 and quantified using ImageQuant software (Cytiva).

## E2–Ub(T) to E3[HECT] pulse–chase Ub thioester transfer assays

For E2–Ub(T) to E3[HECT]-Ub thioester transfer assays, for each replicate, indicated mutational variants or wild-type Ubc4 (7 µM) were charged with wild type or indicated mutational variant of fluorescein-Ub (12 µM) using E1 (300 nM) in 20 mM sodium HEPES pH 7.5, 100 mM NaCl, 5 mM MgCl$_2$, 2 mM ATP, 0.1 mM TCEP in 40 µl reaction volume. After incubation for 30 min at 25 °C, reactions were transferred to ice/water bath and E1 activity was quenched by fourfold dilution with 20 mM sodium HEPES pH 7.5, 100 mM NaCl, 40 mM EDTA. Subsequent steps were performed on ice/water bath. Mixtures were diluted with 20 mM sodium HEPES pH 7.5, 100 mM NaCl, 0.1 mM TCEP, 0.1 mg ml⁻¹ ovalbumin to a final E2-Ub(T) concentration of 30 nM. A control without E3[HECT] was obtained by diluting a 100 µl aliquot with an equal volume of 20 mM sodium HEPES pH 7.5, 100 mM NaCl, 0.1 mM TCEP, and 0.1 mg ml⁻¹ ovalbumin. Thioester transfer reactions (chase) were initiated by diluting a 100 µl aliquot with an equal volume of a solution of a specific variant of Pub2[HECT] at 40 nM in 20 mM sodium HEPES pH 7.5, 100 mM NaCl, 0.1 mM TCEP, and 0.1 mg ml⁻¹ ovalbumin. Concentrations at time 0 were 15 nM for fluorescein-Ub-E2 and 20 nM for E3[HECT]. Aliquots (40 µl) were removed at indicated timepoints and quenched by addition of LDS NuPAGE buffer (final concentration of 1×). Products were separated on 4–12% NuPAGE BIS-Tris gels with MOPS running buffer (Thermo Fisher Scientific). To increase fluorescence signal by converting fluorescein to its di-anionic form, gels were incubated in 50 mM Tris•HCl pH 9.5 for 3 min prior to scanning using Amersham Typhoon 5 (Cytiva). Bands were quantified using ImageQuant software (Cytiva).

## Statistics and reproducibility

Generation of the E1–Ub(T)–E2 and E2–Ub(T)–E3 transthiolation analogues for structural analysis was reproduced with at least three independent purifications. All biochemical experiments were replicated three times, reproduced with at least two independent purifications, and reproduced independently at least twice. Statistical analyses and graphing of the data were performed using Prism 10.1.0 (GraphPad Software). Number of replicates and details on statistical analyses and test are provided in the methods pertaining to each experiment and/or the appropriate legend.

## Reporting summary

Further information on research design is available in the Nature Portfolio Reporting Summary linked to this article.

## Data availability

Cryo-EM reconstructions and coordinates are deposited and are available at the Electron Microscopy Data Bank (http://emdataresource.org/) and PDB (https://www.rcsb.org/), respectively. For singly loaded E1–Ub(T)–E2, cryo-EM coordinates and maps deposited under accession codes 9B5M and EMD-44217 (consensus), 9B5N and EMD-44218 (consensus, state 1), 9B5O and EMD-44219 (consensus, state 10), 9B5P and EMD-44220 (cluster 1), 9B5U and EMD-44225 (cluster 1, state 1), 9B5V and EMD-44226 (cluster 1, state 10), 9B5Q and EMD-44221 (cluster 2), 9B5R and EMD-44222 (cluster 3), 9B5S and EMD-44223 (cluster 4), 9B5T and EMD-44224 (cluster 5), 9B5W and EMD-44227 (cluster 5, state 1) and 9B5X and EMD-44228 (cluster 5, state 10) with cryo-EM maps for 3D classes representing states 1–10 as additional maps under accession codes for consensus (10 maps) and for clusters 1–5 (50 maps, 10 per cluster). For doubly loaded E1–Ub(T)–E2 with Ub(A), cryo-EM coordinates and maps are deposited under accession codes 9B5C and EMD-44207 (consensus), 9B5D and EMD-44208 (consensus, state 1), 9B5E and EMD-44209 (consensus, state 10), 9B5F and EMD-44210 (cluster 1), 9B5K and EMD-44215 (cluster 1, state 1), 9B5G and EMD-44211 (cluster 2), 9B5H and EMD-44212 (cluster 3), 9B5I and EMD-44213 (cluster 4), 9B5J and EMD-44214 (cluster 5), and 9B5L and EMD-44216 (cluster 5, state 10) with cryo-EM maps for 3D classes representing states 1–10 as additional maps under accession codes for consensus (10 maps) and for clusters 1–5 (50 maps, 10 per cluster). For E2–Ub(T)–E3, cryo-EM coordinates and maps are deposited under accession codes 9B55 and EMD-44200 (state 1), 9B56 and EMD-44201 (state 2), 9B57 and EMD-44202 (state 3), 9B58 and EMD-44203 (state 4), 9B59 and EMD-44204 (state 5), 9B5A and EMD-44205 (state 6), and 9B5B and EMD-44206 (state 7). All relevant data are included in the manuscript. There are no restrictions on data availability. Source data are provided with this paper.

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

**Acknowledgements** The authors thank members of the Lima laboratory for advice; M. J. de la Cruz for assistance during cryo-EM data collection; and N. H. Jones for initial stages involving cloning, purification and assays for the Pub2 HECT domain. This research was supported in part by NIH grants R35 GM118080 (C.D.L.), a NCI Cancer Center support grant (P30 CA008748), T32 CA062948 (A.M.V.D.R.), T32 GM115327 (M.C.L.) and R01 AI118224 (D.S.T.). The content is solely the responsibility of the authors and does not represent the official views of the National Institutes of Health. C.D.L. is an investigator of the Howard Hughes Medical Institute. For the purpose of open access, the authors have applied a CC BY public copyright license to any author accepted manuscript version arising from this submission.

**Author contributions** C.D.L. and D.S.T. conceived the project. T.K. isolated and characterized E1–Ub(T)–E2 and E2–Ub(T)–E3^HECT complexes using cryo-EM, and conducted biochemical analyses and all other experiments. T.K. and C.D.L. interpreted cryo-EM data and produced atomic models. Z.S.H. contributed to establishing the method of Ub-PSAN semisynthesis and performed an initial assessment of the probe reactivity. M.C.L., A.M.V.D.R., C.J. and D.S.T. contributed to development of $H_2N$-Gly–PSAN. M.C.L., A.M.V.D.R. provided $H_2N$-Gly–PSAN. T.K. and C.D.L. analysed the data, generated figures and wrote the manuscript with input from all authors.

**Competing interests** The authors declare no competing interests.

**Additional information**
**Correspondence and requests for materials** should be addressed to Derek S. Tan or Christopher D. Lima.

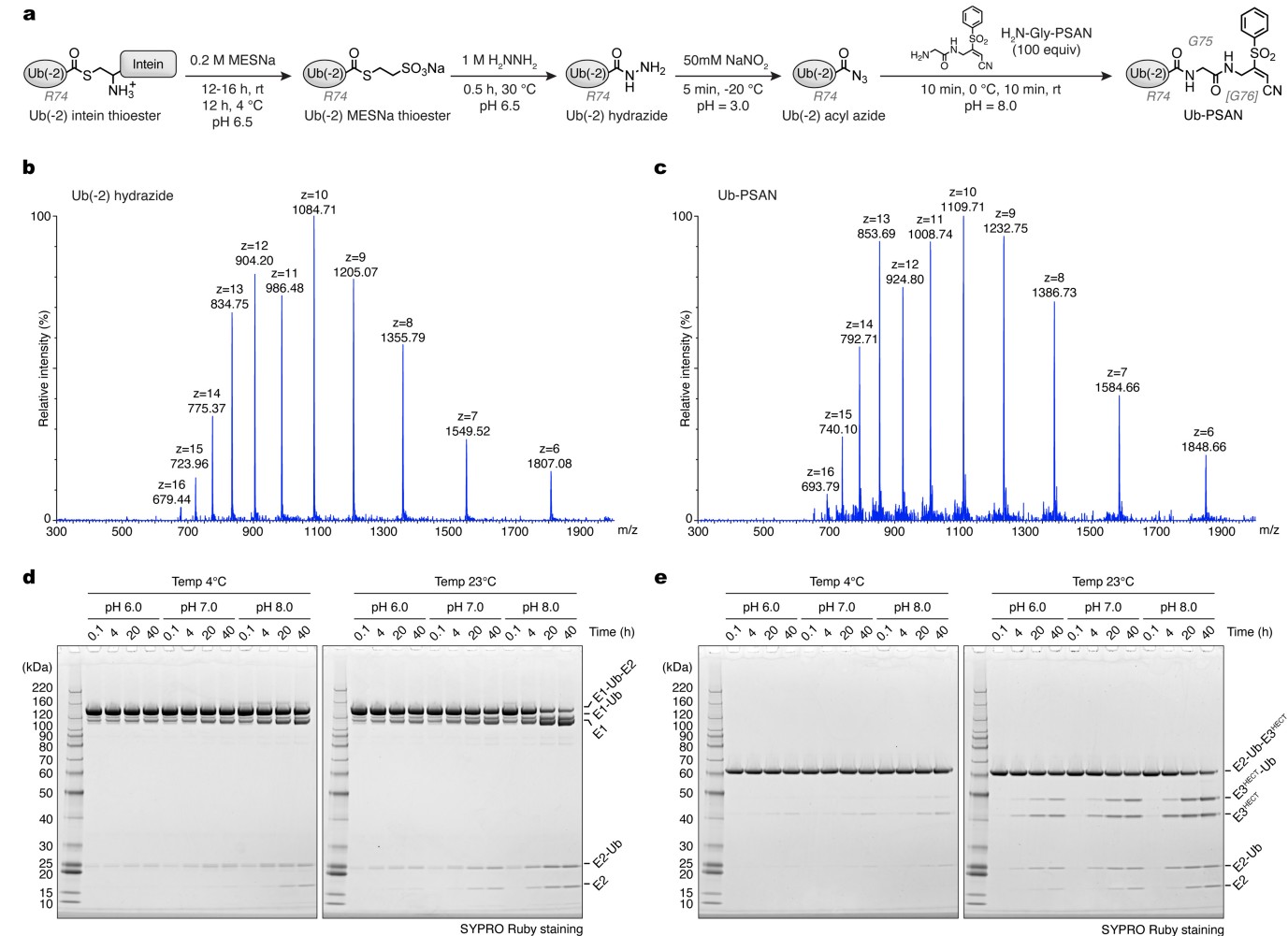

**Extended Data Fig. 1 | Semisynthesis of Ub-PSAN and compositional stability of the E1-Ub(T)-E2 and E2-Ub(T)-E3^HECT transthiolation analogues. a**, Schematic of semisynthesis of Ub-PSAN from a Ub(-2) intein fusion protein expressed in *E. coli* (shown here as thioester intermediate following intein-catalyzed N-to-S acyl transfer), and synthetic H₂N-Gly-PSAN (3-[phenylsulfonyl] acrylonitrile). Ub(-2) is truncated by last two glycine residues at its C-terminus and contains StrepTag and TEV protease cleavage site (MWSHPQFEKSAENLYFQGSGG) added to its N-terminus. Positions of Ub residues R74 and G75 in italics, PSAN containing surrogate of G76 additionally indicated in brackets. **b,c**, Electrospray ionization-mass spectrometry (ESI-MS) spectra of Ub(-2) hydrazide (mass observed, 10840.5 Da; theoretical, 10839.3 Da) (**b**), and Ub-PSAN (mass observed, 11085.5 Da; theoretical, 10886.6 Da) (**c**). **d,e**, SDS-PAGE analysis of the purified E1-Ub(T)-E2 complex (**d**), and purified E2-Ub(T)-E3^HECT complex (**e**), incubated at indicated pH and temperatures. Gels are representative of three independent preparations of protein complexes.

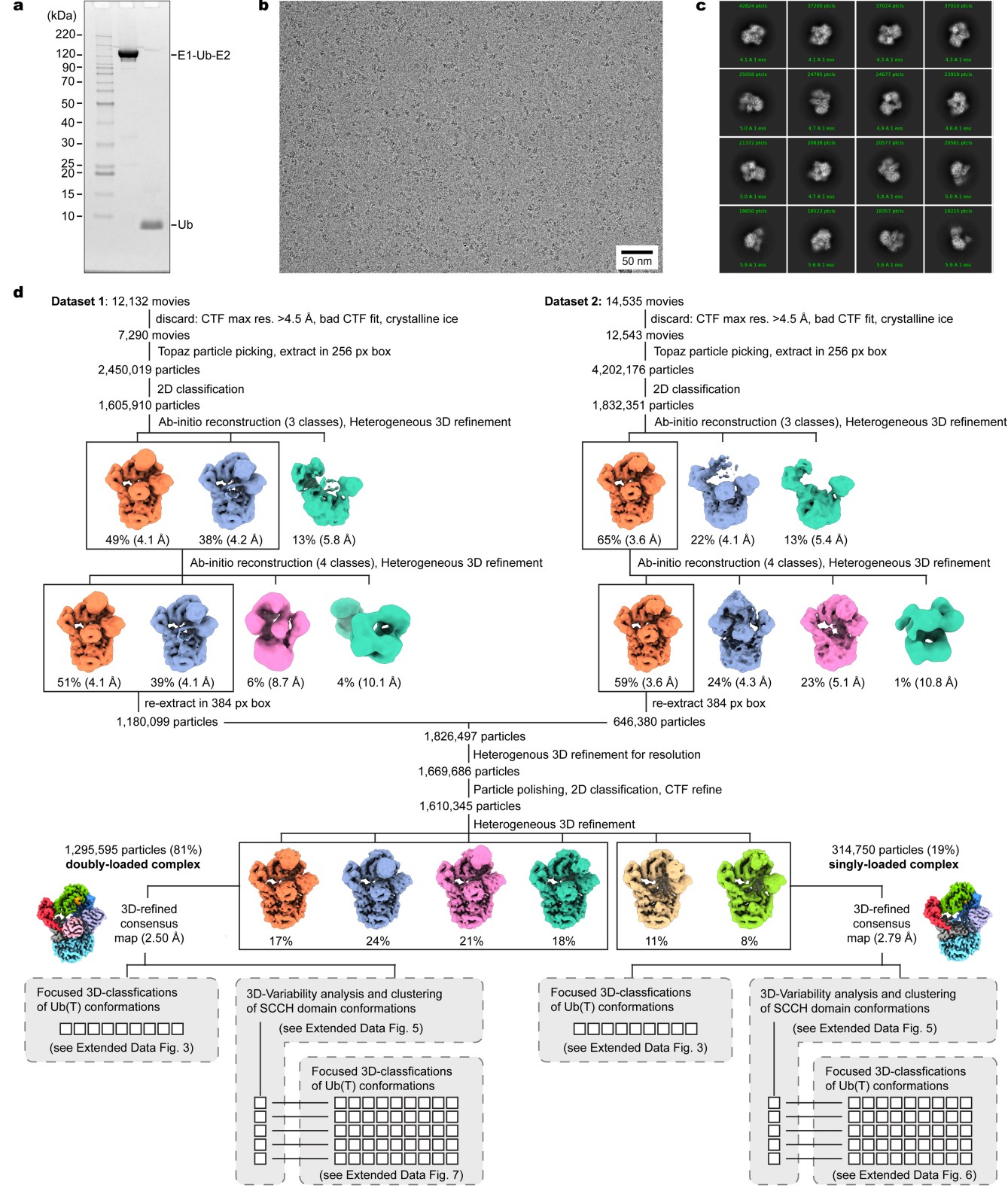

**Extended Data Fig. 2 | Cryo-EM data collection and processing of E1-Ub(T)-E2 complexes. a**, SDS-PAGE analysis of sample used in cryo-EM sample preparation representative of three independent preparations. **b**, Representative micrograph (out of 19,833 images used for particle picking), scale bar = 50 nm. **c**, Representative 2D-class averages from initial reference-free 2D classification (representative of 200 classes), **d**, Flowchart of the image processing of the cryo-EM data. Resolutions indicated in parentheses.

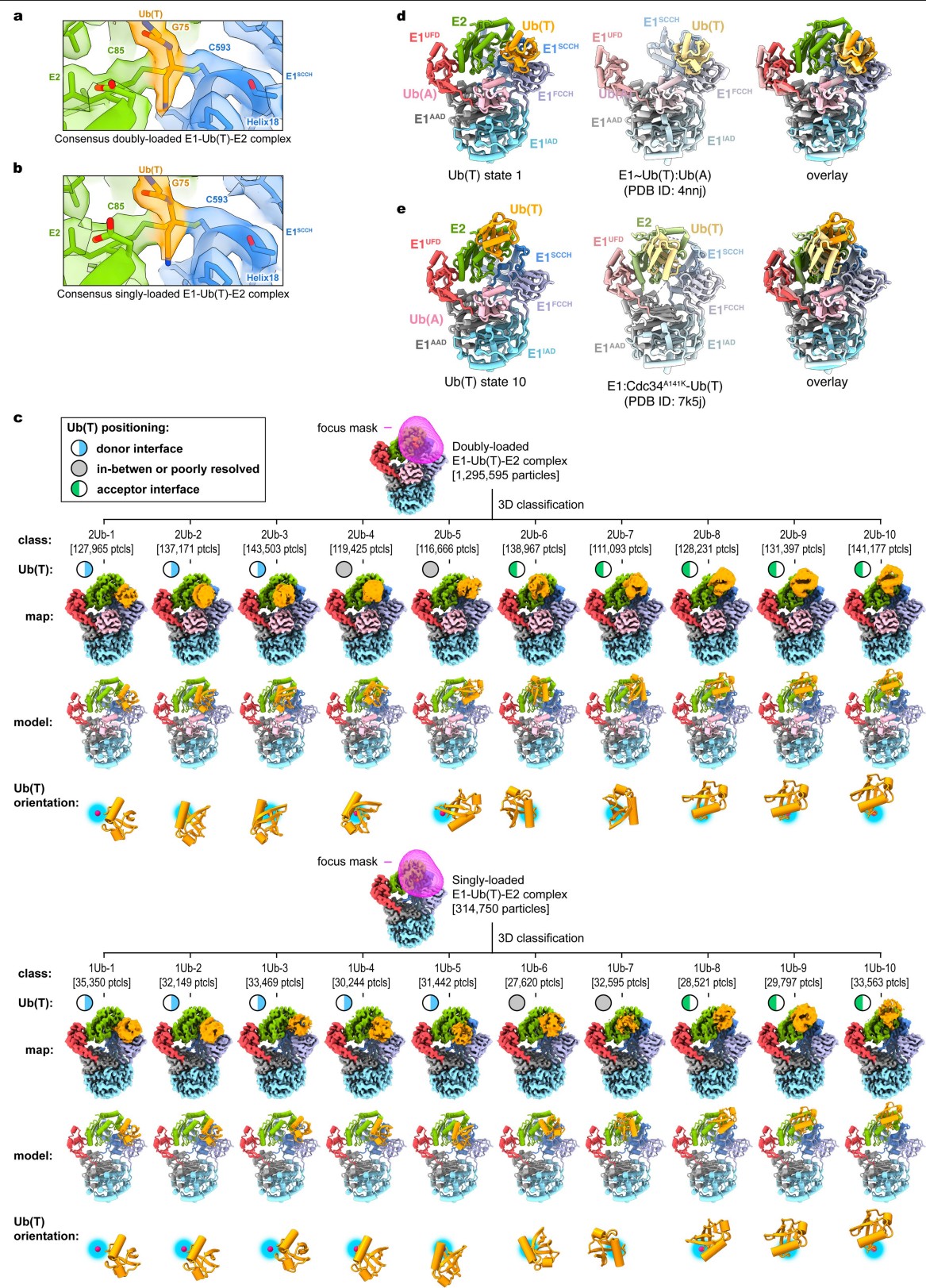

**Extended Data Fig. 3 | Consensus EM reconstructions and focused Ub(T) classifications in E1-Ub(T)-E2 complexes. a**,**b**, EM densities and the fitted atomic models of the transthiolation site in the consensus reconstructions of E1-Ub(T)-E2:Ub(A) (doubly-loaded) complex (**a**) and E1-Ub(T)-E2 (singly-loaded) complex (**b**). **c**, Reconstructions and corresponding models of Ub(T) states after 3D classification without image realignment, performed with a focus mask on Ub(T) in doubly- and singly-loaded complexes. **d**, Comparison of doubly-loaded E1-Ub(T)-E2 complex (State 1) to doubly-loaded E1-Ub(T) in the absence of E2 (PDB ID: 4NNJ) with Ub(T) interacting with the FCCH domain of E1. **e**, Comparison of doubly-loaded E1-Ub(T)-E2 complex (State 10) to singly-loaded E1-Ub(T)-E2 where E2 is Cdc34 (PDB ID: 7K5J) with Ub(T) in contact with E2. Isosurface levels contoured at 0.67, 0.65, 0.5 (**a**,**b**,**c**).

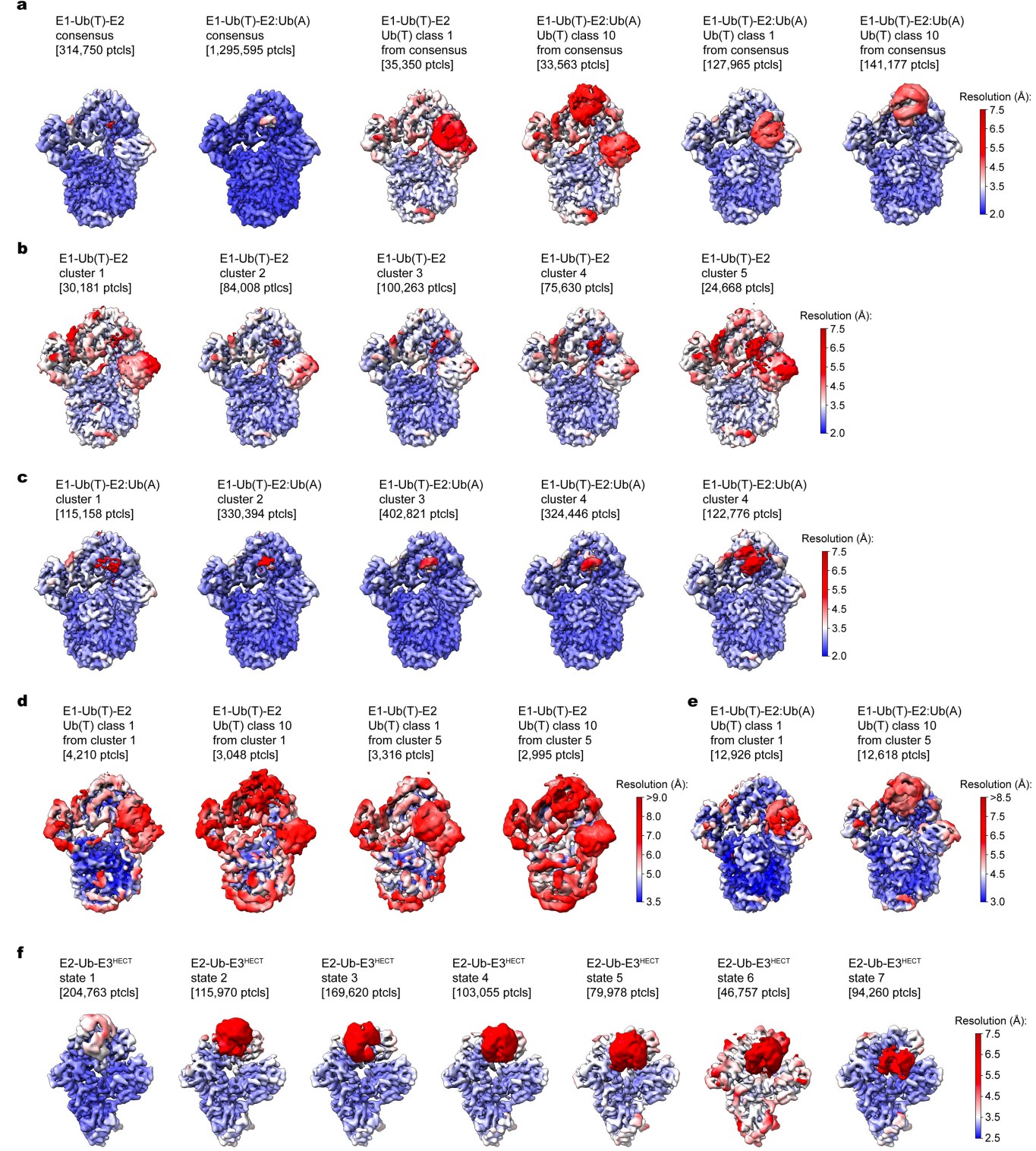

**a**

E1-Ub(T)-E2
consensus
[314,750 ptcls]

E1-Ub(T)-E2:Ub(A)
consensus
[1,295,595 ptcls]

E1-Ub(T)-E2
Ub(T) class 1
from consensus
[35,350 ptcls]

E1-Ub(T)-E2:Ub(A)
Ub(T) class 10
from consensus
[33,563 ptcls]

E1-Ub(T)-E2:Ub(A)
Ub(T) class 1
from consensus
[127,965 ptcls]

E1-Ub(T)-E2:Ub(A)
Ub(T) class 10
from consensus
[141,177 ptcls]

**b**

E1-Ub(T)-E2
cluster 1
[30,181 ptlcs]

E1-Ub(T)-E2
cluster 2
[84,008 ptlcs]

E1-Ub(T)-E2
cluster 3
[100,263 ptlcs]

E1-Ub(T)-E2
cluster 4
[75,630 ptlcs]

E1-Ub(T)-E2
cluster 5
[24,668 ptls]

**c**

E1-Ub(T)-E2:Ub(A)
cluster 1
[115,158 ptcls]

E1-Ub(T)-E2:Ub(A)
cluster 2
[330,394 ptcls]

E1-Ub(T)-E2:Ub(A)
cluster 3
[402,821 ptcls]

E1-Ub(T)-E2:Ub(A)
cluster 4
[324,446 ptcls]

E1-Ub(T)-E2:Ub(A)
cluster 4
[122,776 ptcls]

**d**

E1-Ub(T)-E2
Ub(T) class 1
from cluster 1
[4,210 ptcls]

E1-Ub(T)-E2
Ub(T) class 10
from cluster 1
[3,048 ptcls]

E1-Ub(T)-E2
Ub(T) class 1
from cluster 5
[3,316 ptcls]

E1-Ub(T)-E2
Ub(T) class 10
from cluster 5
[2,995 ptcls]

**e**

E1-Ub(T)-E2:Ub(A)
Ub(T) class 1
from cluster 1
[12,926 ptcls]

E1-Ub(T)-E2:Ub(A)
Ub(T) class 10
from cluster 5
[12,618 ptcls]

**f**

E2-Ub-E3HECT
state 1
[204,763 ptcls]

E2-Ub-E3HECT
state 2
[115,970 ptcls]

E2-Ub-E3HECT
state 3
[169,620 ptcls]

E2-Ub-E3HECT
state 4
[103,055 ptcls]

E2-Ub-E3HECT
state 5
[79,978 ptcls]

E2-Ub-E3HECT
state 6
[46,757 ptcls]

E2-Ub-E3HECT
state 7
[94,260 ptcls]

**Extended Data Fig. 4 | Local resolution estimates for consensus, singly- and doubly loaded E1-Ub(T)-E2 and E2-Ub(T)-E3 cryo-EM reconstructions.** Names and corresponding particle counts (shown in brackets) are specified above each reconstruction along with a scale bar indicating resolution range to the right of each row where that scale applies. **a**, Reconstructions for consensus doubly and singly-loaded E1-Ub(T)-E2 complexes and corresponding Ub(T) donor and acceptor classes. **b**, Reconstructions for five conformational clusters of SCCH domain rotation in singly-loaded E1-Ub(T)-E2 complex. **c**, Reconstructions for five conformational clusters of SCCH domain rotation in doubly-loaded E1-Ub(T)-E2 complex. **d**,**e**, Reconstructions for Ub(T) donor and acceptor classes in cluster 1 and 5 of SCCH domain rotation in singly- (**d**) and doubly-loaded (**e**) E1-Ub(T)-E2 complexes. **f**, Reconstructions for states 1 through 7 of E2-Ub(T)-E3HECT. Isosurface levels contoured at 0.48, 0.48, 0.50, 0.47, 0.48, 0.48 (left to right in **a**), 0.48 (all maps in **b**), 0.48, 0.46, 0.47, 0.46, 0.47 (left to right in **c**), 0.53, 0.47, 0.53, 0.48 (left to right in **d**), 0.54, 0.55 (left to right in **e**), 0.48 (all maps in **b**).

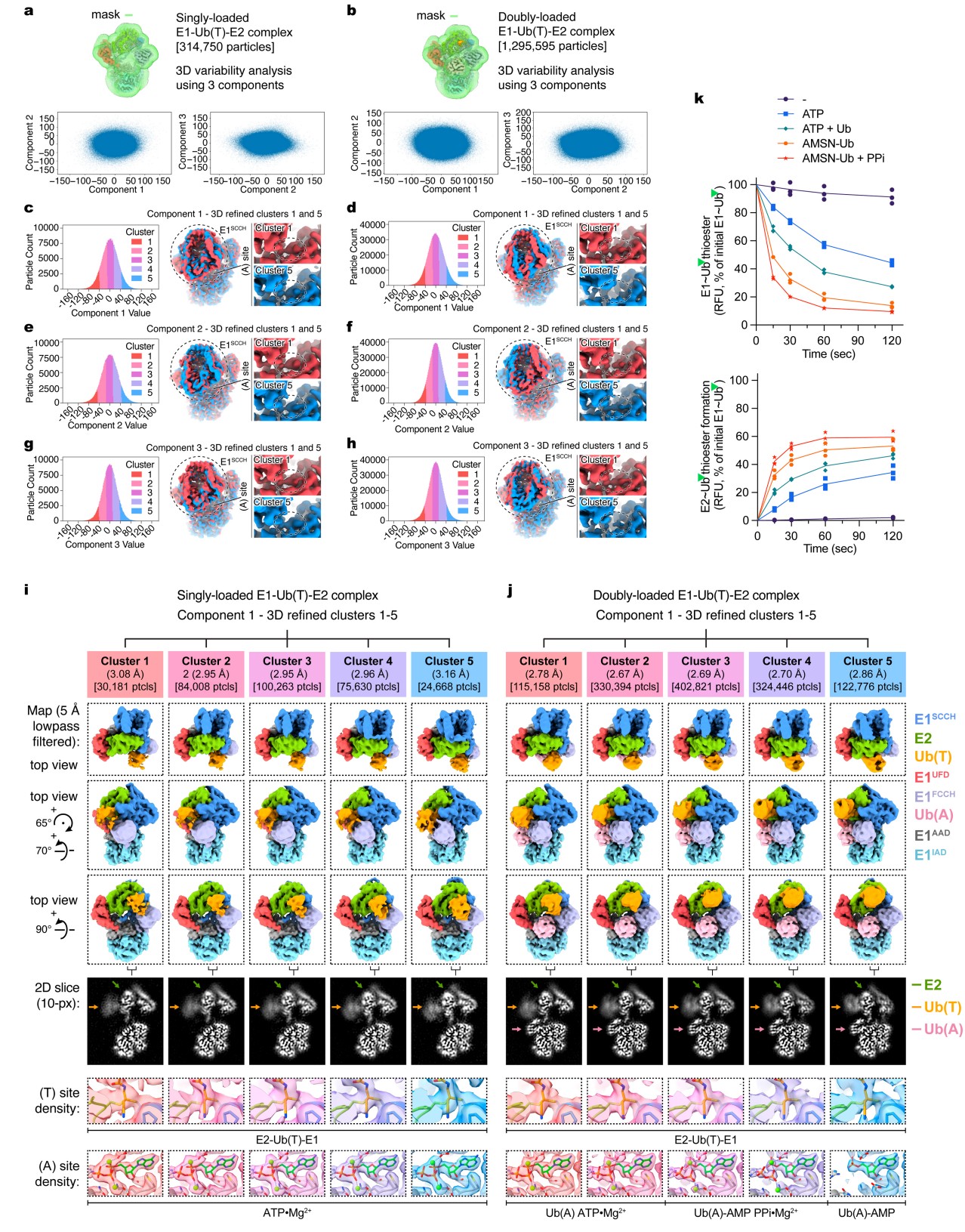

**Extended Data Fig. 5 |** See next page for caption.

**Extended Data Fig. 5 | 3D variability analysis to resolve the rotation of the SCCH domain in singly and doubly-loaded E1-Ub(T)-E2 complex and transthiolation assays.** 3D variability analysis for singly-loaded (**a**) and doubly loaded (**b**) E1 performed with a mask on E1-E2, excluding Ub(T), as detailed in Methods. 2D scatter plots show particle latent coordinates of sequential pairs of variability components. **c**,**e**,**g**, Histograms of variability components for singly-loaded complex with binning of the particles into 5 non-overlapping clusters. Maps from 3D refinements of clusters 1 (red) and 5 (blue) for each component shown next to magnified views of the adenylation (A) site. Component 1 (**c**) captures the largest range of E1$^{SCCH}$ domain orientations as evidenced by comparing overlap between red and blue maps. **d**,**f**,**h**, Histograms of variability components obtained for doubly-loaded complex showing binning of the particles into 5 non-overlapping clusters. Maps from 3D refinements of clusters 1 (red) and 5 (blue) for each component shown next to magnified views of the adenylation (A) site. Component 1 (**d**) captures the largest range of E1$^{SCCH}$ domain orientations as evidenced by comparing overlap between red and blue maps. **i**,**j**, detailed analysis of all reconstructions obtained for clusters 1–5 shown in panels **c**,**d**. Resolution of reconstructions indicated in parentheses. Maps were lowpass filtered at 5 Å to visualize conformational differences in complexes shown in three views. The 2D slice view shows corresponding summed 10-pixel (10.64 Å thick) cross-sections of the map at the nominal resolution. Views of the adenylation site ((A) site) and transthiolation site ((T) site) show EM densities at the nominal resolution of the map with fitted atomic models for the indicated cluster. Isosurface levels contoured at 0.25-0.28 (low pass filtered maps) (**i**); 0.41, 0.43, 0.41, 0.52, 0.44 for clusters 1, 2, 3, 4, 5 in singly-loaded complex and 0.66, 0.59, 0.58, 0.48, 0.46 for clusters 1, 2, 3, 4, 5 in doubly-loaded complex ((T) site); 1.1, 1.1, 1.0, 1.0, 0.9 for clusters 1, 2, 3, 4, 5 in singly-loaded complex and 1.2, 1.2, 1.2, 1.2, 1.2 for clusters 1, 2, 3, 4, 5 in doubly-loaded complex ((A) site). **k**, Quantification of the in-gel fluorescence of the gels presented in Supplementary Fig. 3. Data points for three replicates (symbols) and average (line) is plotted versus reaction time. RFU, relative fluorescence units. Green triangle next to ubiquitin indicates fluorescein.

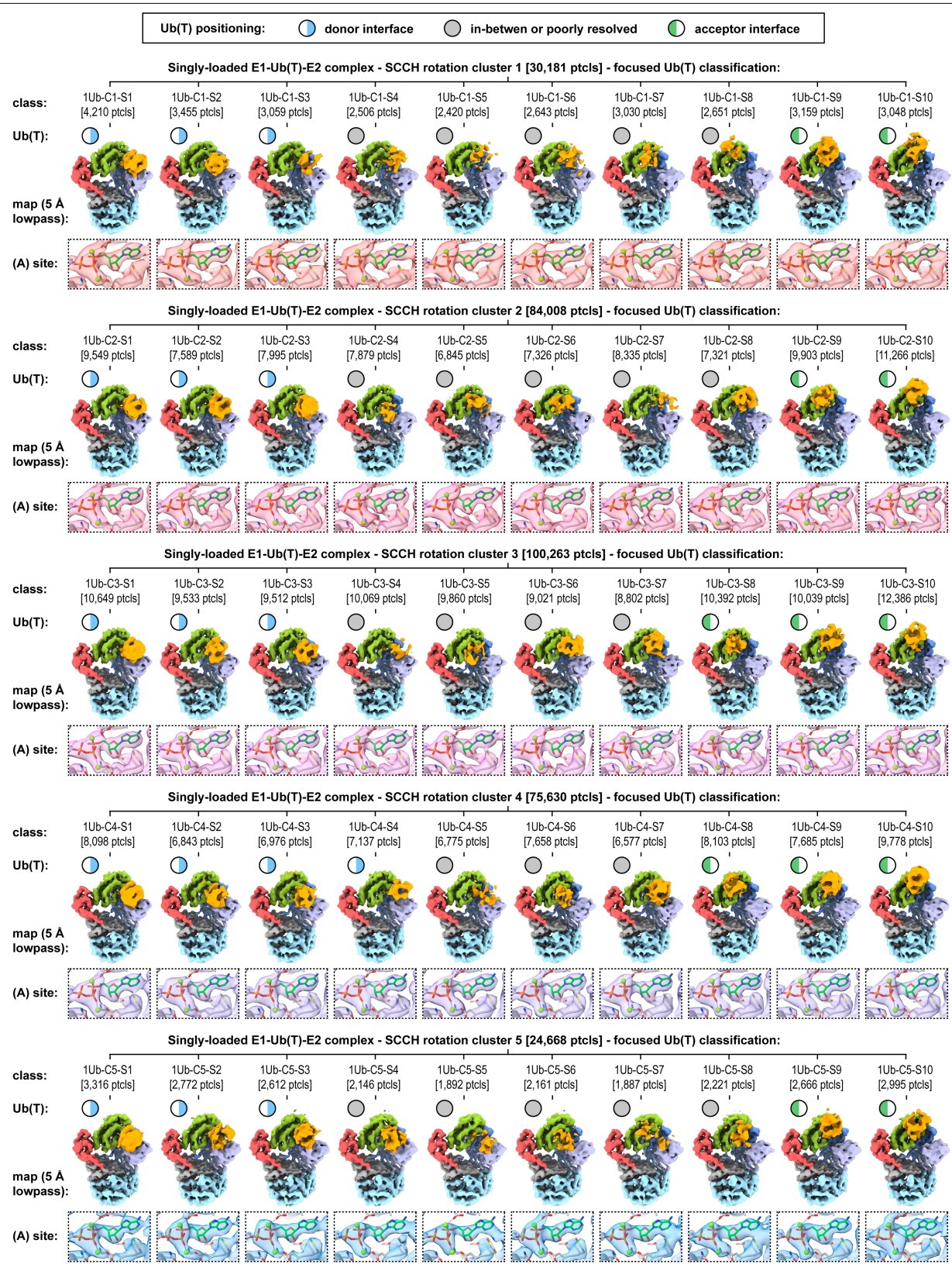

**Extended Data Fig. 6 | Focused Ub(T) classifications in each of five conformational clusters of SCCH domain rotation in singly-loaded E1-Ub(T)-E2 complexes.** Classifications were performed without image realignment using a focus mask on Ub(T), as shown in Extended Data Fig. 3. Overall views show maps lowpass filtered at 5 Å to better visualize Ub(T) density. The views of the adenylation site ((A) site) represent of EM density at the nominal resolution of the map superimposed with the consensus atomic model for the corresponding cluster. Isosurface levels contoured at 0.5 (5 Å lowpass filtered maps) and 0.7 ((A) site).

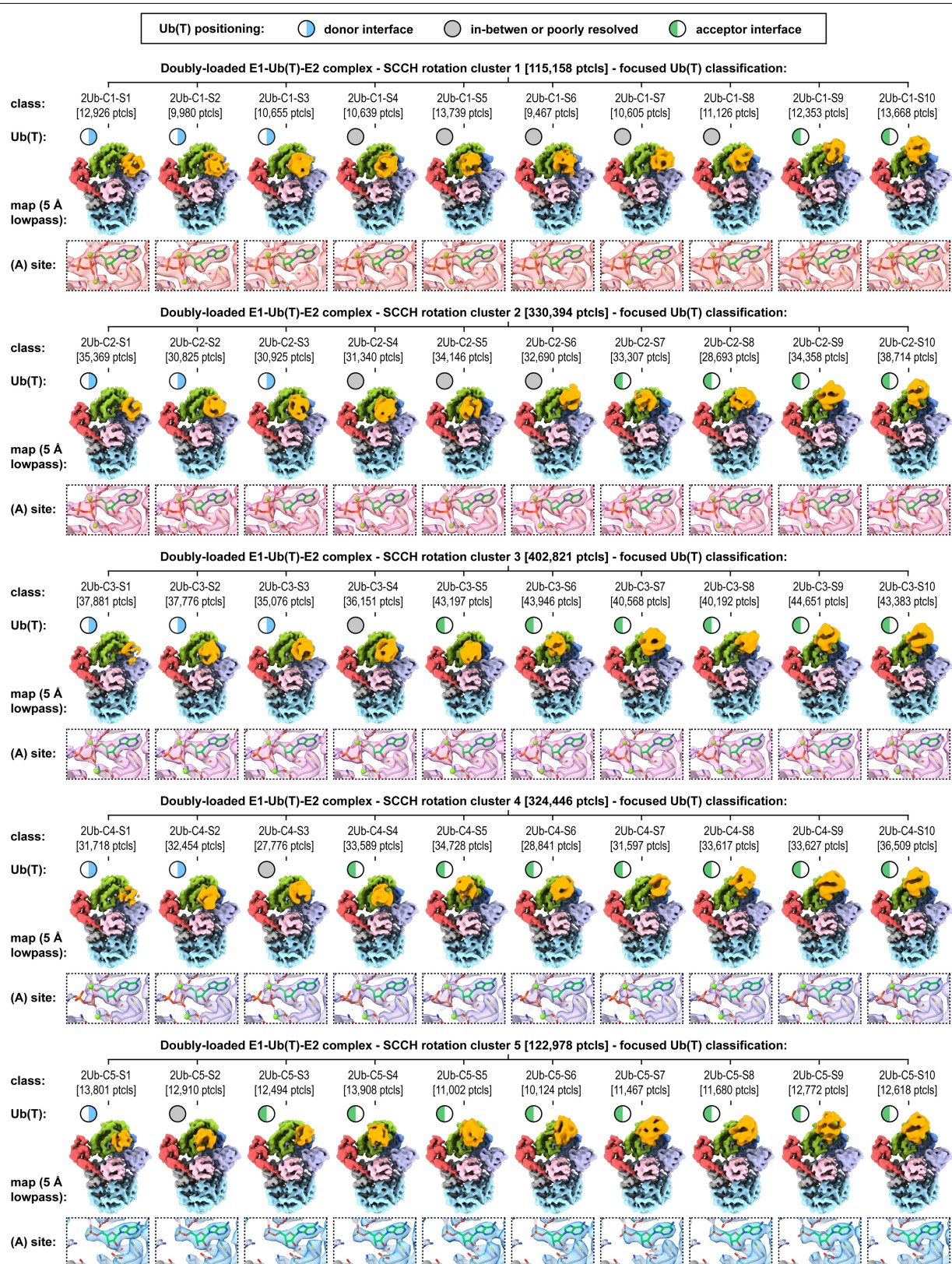

**Extended Data Fig. 7 | Focused Ub(T) classifications in each of five conformational clusters of SCCH domain rotation in doubly-loaded E1-Ub(T)-E2 complexes.** Classifications were performed without image realignment using a focus mask on Ub(T), as shown in Extended Data Fig. 3. Overall views show maps lowpass filtered at 5 Å to better visualize Ub(T) density. The views of the adenylation site ((A) site) represent of EM density at the nominal resolution of the map superimposed with the consensus atomic model for the corresponding cluster. Isosurfaces contoured at level 0.5 (5 Å lowpass filtered maps) and 0.7 ((A) site).

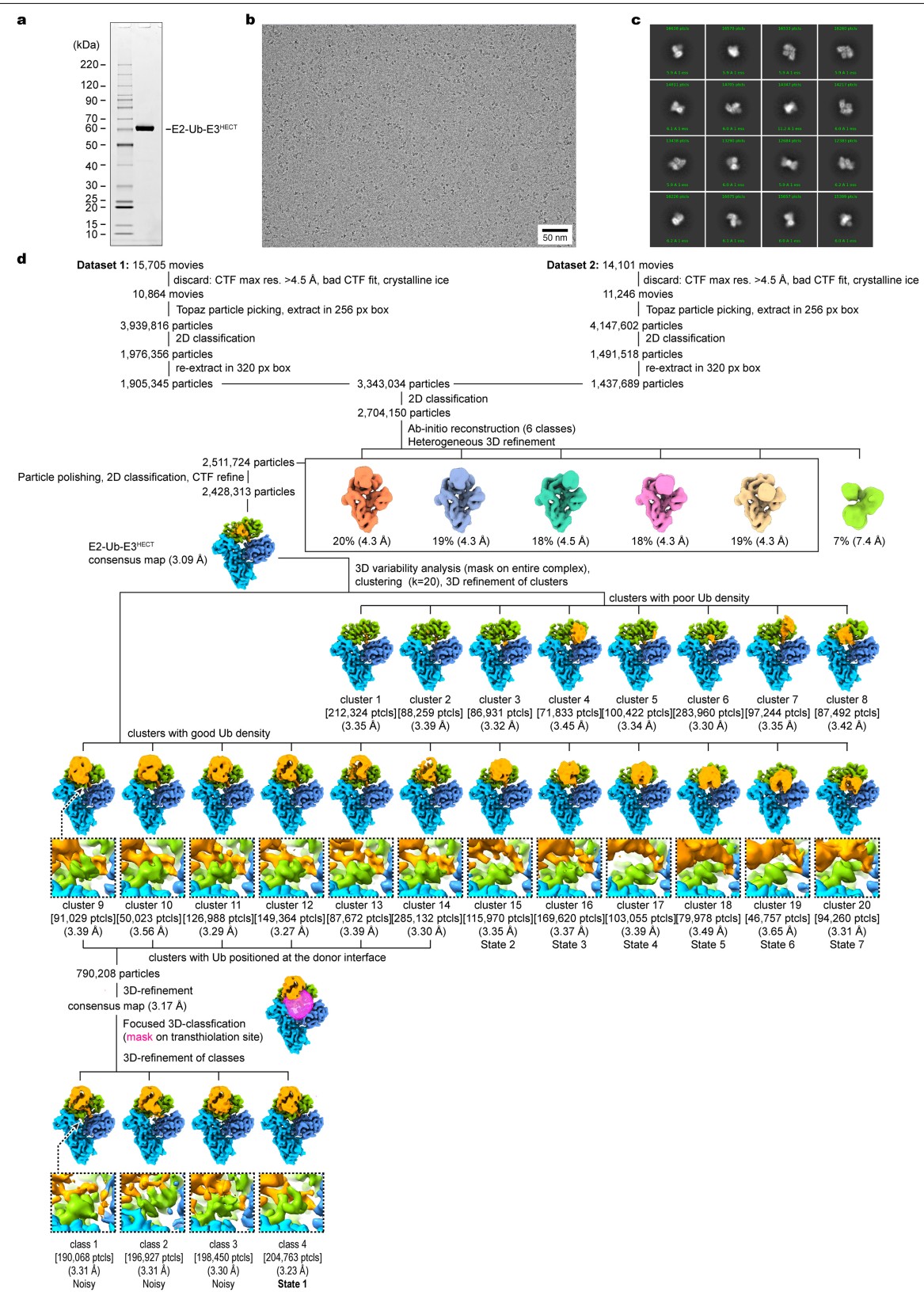

**Extended Data Fig. 8 | Cryo-EM data collection and processing of E2-Ub(T)-E3^HECT to resolve conformational states of the E2-Ub(T)-E3^HECT complex. a**, SDS-PAGE analysis of sample used in cryo-EM sample preparation representative of three independent preparations. **b**, Representative micrograph (out of 22,110 images used for particle picking), Scale bar = 50 nm. **c**, Representative 2D-class averages from initial reference-free 2D classification (representative of 100 classes), **d**, Flowchart of the image processing of the cryo-EM data. The resolutions are indicated in parentheses. Clusters are labeled with number of particles enclosed in brackets and resolution in parentheses. States 1–7 are indicated below the relevant cluster or class.

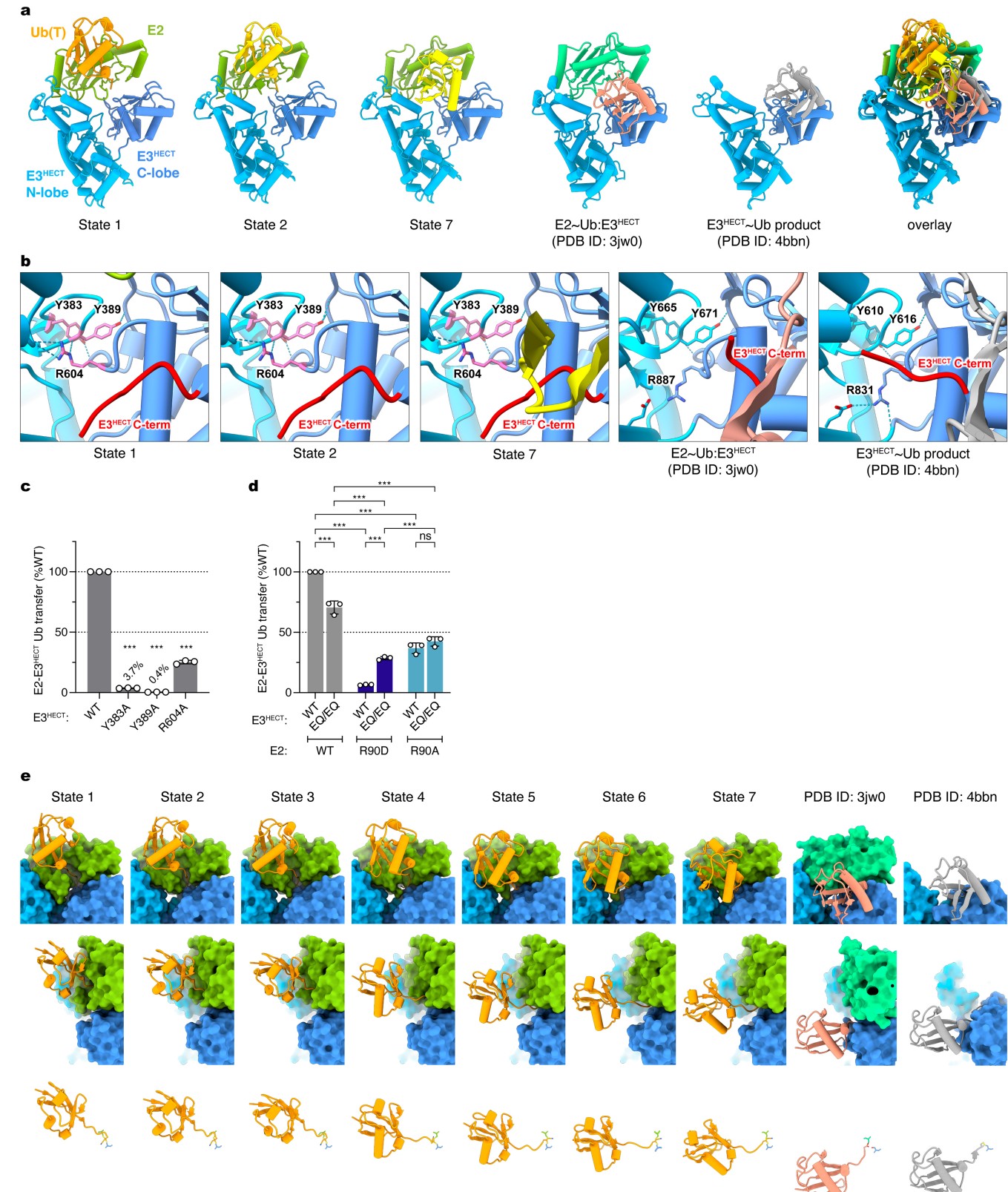

**Extended Data Fig. 9** | See next page for caption.

**Extended Data Fig. 9 | Comparison of E2-Ub(T)-E3$^{HECT}$ in states 1, 2 and 7 (this study) with published structures of E3s$^{HECT}$ and structure-function analysis of the interface between N and C-lobes in E3$^{HECT}$. a**, Comparison of states 1, 2 and 7 of E2-Ub(T)-E3$^{HECT}$ showing similar orientation of the N and C-lobes of E3$^{HECT}$ and distinct positioning and conformation of Ub(T) compared to crystal structures of E2-Ub(T):E3$^{HECT}$ complex (Protein Data Bank (PDB) ID: 3JW0) and E3$^{HECT}$-Ub(T) (PDB ID: 4BBN). **b**, Magnified view of the interface between N and C-lobes in the structures shown in (**a**). **c**, Histograms derived by quantification of the in-gel fluorescence of the gels presented in (Supplementary Fig. 4). Bars represent mean ± s.d of n = 3 replicates. Statistical differences between wild-type and mutants were determined by two-tailed unpaired t-test: ***$P$ < 0.001.

**d**, Histograms derived by quantification of the in-gel fluorescence of the gels presented (Supplementary Fig. 4). EQ/EQ indicate E451Q/E455Q mutation in E3. Bars represent mean ± s.d of n = 3 replicates. Data were analyzed by two-sided one-way ANOVA with Tukey's test: ***$P$ < 0.001, ns, not significant.
**e**, Orthogonal views of surfaces for models for E2 and E3 for states 1 through 7 with ubiquitin shown as in cartoon representation next to similar depictions of PDB ID: 3JW0 and 4BBN to illustrate the rotation and translation of ubiquit in as it transits between states 1–7 to its product conformation in 3JW0 and 4BBN. Cartoon of ubiquitin shown at the bottom aligned at its C-terminus in respective complexes to provide another view of Ub(T) transitioning through states 1–7 compared to Ub(T) in PDB ID: 3JW0 and 4BBN.

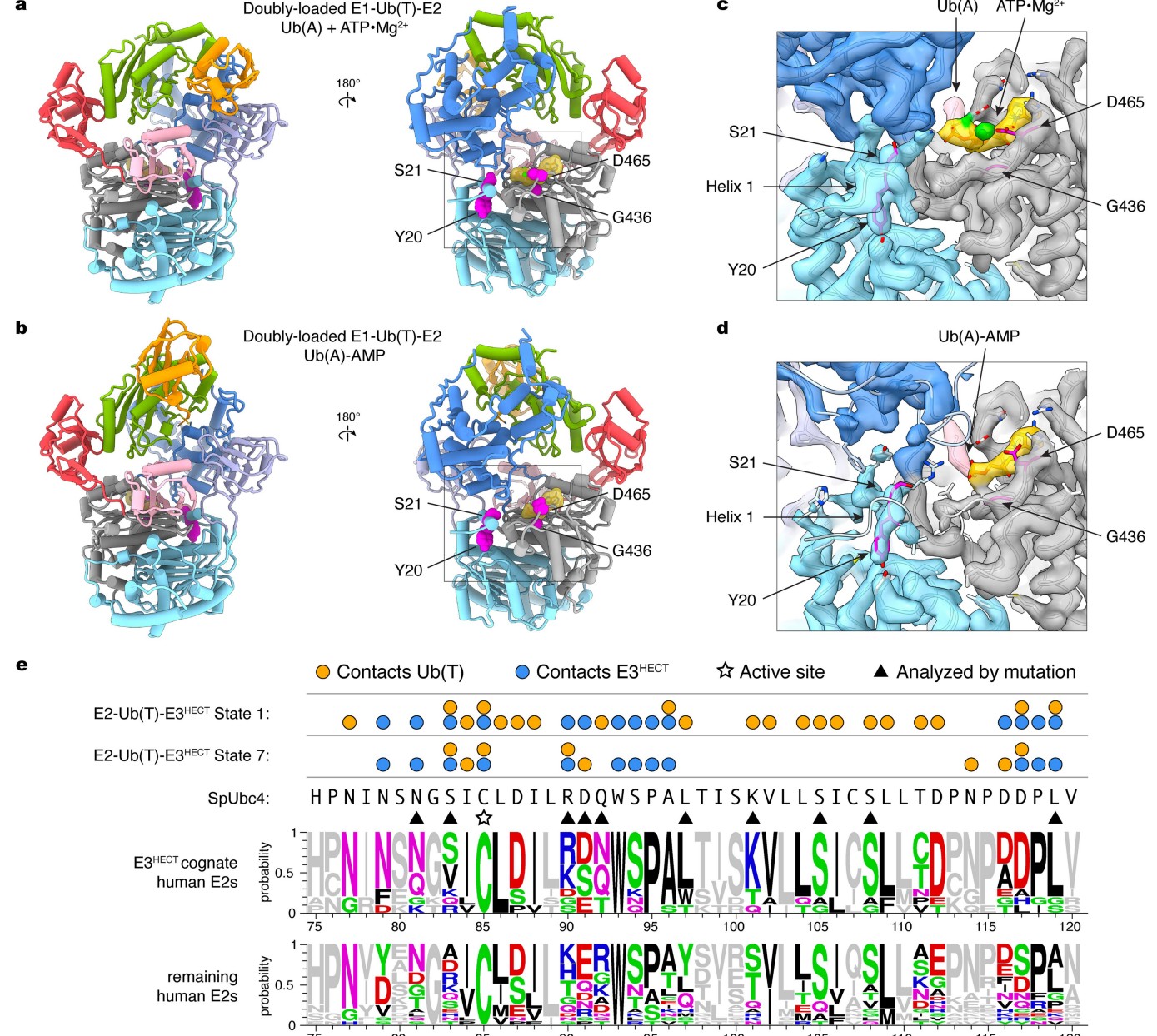

**Extended Data Fig. 10 | Select disease-associated mutations affecting transthiolation in human E1 and sequence conservation of residues important for transthiolation among human E3^HECT cognate E2s.**
**a,b**, Residues corresponding to VEXAS syndrome mutations (VEXAS mutations Y55H, S56F, G477A and D506N/G correspond to residues Y20, S21, G436 and D465 in *S. pombe* Uba1, respectively) affecting E1-E2 transthiolation mapped to onto structures of doubly-loaded E1-Ub(T)-E2 complex (cluster 1 – ATP Mg^2+ bound) with Ub at the donor interface and doubly-loaded E1-Ub(T)-E2 complex (cluster 5 – Ub-AMP bound) shown in magenta and labeled. Model color coded as in Fig. 2. **c,d**, Magnified views showing EM densities and fitted atomic models of the region surrounding adenylation site in doubly-loaded E1-Ub(T)-E2 complex (cluster 1 – ATP Mg^2+ bound) with Ub at the donor interface (**c**) and doubly-loaded E1-Ub(T)-E2 complex (cluster 5 – Ub-AMP bound) shown in magenta and labeled (**d**). Isosurface levels contoured at 0.95 and 0.87 (**c,d**). **e**, Sequence alignment of *S. pombe* Ubc4 and human E2 enzymes. Sequence alignment logos (generated using WebLogo[73]) display the conservation of E2

residues among human E2s that form functional pairs with HECT-type E3 ligases and the remaining human E2s, numbered according to Ubc4. Sequences in the alignment are: UBE2D1, UBE2D2, UBE2D3, UBE2D4, UBE2E1, UBE2E2, UBE2E3, UBE2L3, UBE2J1, UBE2J2 (Uniprot accession codes: P51668, P62837, P61077, Q9Y2X8, P51965, Q96LR5, Q969T4, P68036, Q9Y385, Q8N2K1, respectively) for HECT-cognate E2s, and UBE2A, UBE2B, UBE2C, UBE2G1, UBE2G2, UBE2H, UBE2K, UBE2L6, UBE2N, UBE2O, UBE2Q1, UBE2Q2, UBE2QL1, UBE2R1, UBE2R2, UBE2S, UBE2T, UBE2U, UBE2W, UBE2Z (Uniprot accession codes: P49459, P63146, O00762, P62253, P60604, P62256, P61086, O14933, P61088, Q9C0C9, Q7Z7E8, Q8WVN8, A1L167, P49427, Q712K3, Q16763, Q9NPD8, Q5VVX9, Q96B02, Q9H832, respectively) for the remaining E2s. The Ubc4 sequence is shown above, with residues in States 1 and 7 that make contacts within 4.5 Å of Ub(T) and E3^HECT indicated by orange and blue circles, respectively. The active site cysteine and residues mutated in this study are marked by a star and black triangles, respectively.

Derek S. Tan

# Reporting Summary

## Statistics

For all statistical analyses, confirm that the following items are present in the figure legend, table legend, main text, or Methods section.

| n/a | Confirmed | |
|---|---|---|
| ☐ | ☒ | The exact sample size (*n*) for each experimental group/condition, given as a discrete number and unit of measurement |
| ☐ | ☒ | A statement on whether measurements were taken from distinct samples or whether the same sample was measured repeatedly |
| ☐ | ☒ | The statistical test(s) used AND whether they are one- or two-sided<br>*Only common tests should be described solely by name; describe more complex techniques in the Methods section.* |
| ☒ | ☐ | A description of all covariates tested |
| ☐ | ☒ | A description of any assumptions or corrections, such as tests of normality and adjustment for multiple comparisons |
| ☐ | ☒ | A full description of the statistical parameters including central tendency (e.g. means) or other basic estimates (e.g. regression coefficient) AND variation (e.g. standard deviation) or associated estimates of uncertainty (e.g. confidence intervals) |
| ☐ | ☒ | For null hypothesis testing, the test statistic (e.g. *F*, *t*, *r*) with confidence intervals, effect sizes, degrees of freedom and *P* value noted<br>*Give P values as exact values whenever suitable.* |
| ☒ | ☐ | For Bayesian analysis, information on the choice of priors and Markov chain Monte Carlo settings |
| ☒ | ☐ | For hierarchical and complex designs, identification of the appropriate level for tests and full reporting of outcomes |
| ☒ | ☐ | Estimates of effect sizes (e.g. Cohen's *d*, Pearson's *r*), indicating how they were calculated |

*Our web collection on statistics for biologists contains articles on many of the points above.*

## Software and code

Policy information about availability of computer code

| Data collection | Cryo-EM data collection was performed in SerialEM 3.9.0. |
|---|---|
| Data analysis | Cryo-EM data were processed using Relion 3.1.3 , MotionCor2 1.3.2, gctf 1.06, Topaz 0.2.5, cryoSPARC versions 3.3.1 and 4.0.2. Molecular models were docked in UCSF Chimera 1.15 and build and refined using Coot 0.9.8 and PHENIX 1.20.1-4487, and analyzed using MolProbity which is integrated into PHENIX 1.20.1-4487.<br>Structure representations were generated using UCSF ChimeraX 1.7.1. 2D slice views of EM-maps were visualized using IMOD 4.11.<br>Densitometric analysis and visualisation of gel scans was performed in ImageQuant TL versions 8.2.0 and 10.2. Calculations were performed in Excel 16.57. Statistical analyses and plotting of the data was performed in Prism 10.2.0. Sequence alignment logos generated using WebLogo 2.8.2. |

For manuscripts utilizing custom algorithms or software that are central to the research but not yet described in published literature, software must be made available to editors and reviewers. We strongly encourage code deposition in a community repository (e.g. GitHub). See the Nature Portfolio guidelines for submitting code & software for further information.

## Data

Policy information about availability of data

All manuscripts must include a data availability statement. This statement should provide the following information, where applicable:

- Accession codes, unique identifiers, or web links for publicly available datasets
- A description of any restrictions on data availability
- For clinical datasets or third party data, please ensure that the statement adheres to our policy

Cryo-EM reconstructions and coordinates are deposited and available at the Electron Microscopy Data Bank (emdataresource.org) and PDB (rcsb.org), respectively. For singly loaded E1-Ub(T)-E2, cryo-EM coordinates and maps are deposited under accession codes 9B5M and EMD-44217 (consensus), 9B5N and EMD-44218 (consensus, state 1),  9B5O and EMD-44219 (consensus, state 10), 9B5P and EMD-44220 (cluster 1), 9B5U and EMD-44225 (cluster 1, state 1), 9B5V and EMD-44226 (cluster 1, state 10), 9B5Q and EMD-44221 (cluster 2), 9B5R and EMD-44222 (cluster 3), 9B5S and EMD-44223 (cluster 4), 9B5T and EMD-44224 (cluster 5), 9B5W and EMD-44227 (cluster 5, state 1) and 9B5X and EMD-44228 (cluster 5, state 10) with cryo-EM maps for 3D classes representing states 1-10 as additional maps under accession codes for consensus (10 maps) and for clusters 1-5 (50 maps, 10 per cluster). For doubly loaded E1-Ub(T)-E2 with Ub(A), cryo-EM coordinates and maps are deposited under accession codes 9B5C and EMD-44207 (consensus), 9B5D and EMD-44208 (consensus, state 1),  9B5E and EMD-44209 (consensus, state 10), 9B5F and EMD-44210 (cluster 1), 9B5K and EMD-44215 (cluster 1, state 1), 9B5G and EMD-44211 (cluster 2), 9B5H and EMD-44212 (cluster 3), 9B5I and EMD-44213 (cluster 4), 9B5J and EMD-44214 (cluster 5), and 9B5L and EMD-44216 (cluster 5, state 10) with cryo-EM maps for 3D classes representing states 1-10 as additional maps under accession codes for consensus (10 maps) and for clusters 1-5 (50 maps, 10 per cluster). For E2-Ub(T)-E3, cryo-EM coordinates and maps are deposited under accession codes 9B55 and EMD-44200 (state 1), 9B56 and EMD-44201 (state 2), 9B57 and EMD-44202 (state 3), 9B58 and EMD-44203 (state 4), 9B59 and EMD-44204 (state 5), 9B5A and EMD-44205 (state 6), and 9B5B and EMD-44206 (state 7). The atomic coordinates of previously published structures of ubiquitin E1 crosslinked toE2 (Ubc4) with ubiquitin (4II2), ubiquitin E1 bound to ubiquitin-AMSN (6o82) and E2~ubiquitin~HECT (3jw0) were used in this study. All relevant data are included in the manuscript. Supplemental Data Figures contain uncropped gel images for all replicates. There are no restrictions on data availability. Source data are provided with this paper.

## Research involving human participants, their data, or biological material

Policy information about studies with human participants or human data. See also policy information about sex, gender (identity/presentation), and sexual orientation and race, ethnicity and racism.

| | |
|---|---|
| Reporting on sex and gender | n/a |
| Reporting on race, ethnicity, or other socially relevant groupings | n/a |
| Population characteristics | n/a |
| Recruitment | n/a |
| Ethics oversight | n/a |

Note that full information on the approval of the study protocol must also be provided in the manuscript.

# Field-specific reporting

Please select the one below that is the best fit for your research. If you are not sure, read the appropriate sections before making your selection.

☒ Life sciences ☐ Behavioural & social sciences ☐ Ecological, evolutionary & environmental sciences

For a reference copy of the document with all sections, see nature.com/documents/nr-reporting-summary-flat.pdf

# Life sciences study design

All studies must disclose on these points even when the disclosure is negative.

| | |
|---|---|
| Sample size | Sample sizes were not predetermined. Cryo-EM sample size was determined by the available microscope time. The number of images collected is indicated in Extended Data Figures 2 and 8. Biochemical sample sizes were not predetermined. Biochemical sample size was determined after three independent replicates and evaluation of statistical significance to ensure reproducibility. |
| Data exclusions | Cryo-EM images were excluded from the data sets when they showed evidence of crystalline ice, estimated resolution limits worse than 4.5 Å or poor CTF fit. Particles belonging to bad classes were excluded during 2D classifications if their 2D class averages represented noise. The selection of particles are shown in Extended Data Figures 2 and 8. |
| Replication | Cryo-EM: Each condition (E1-Ub(T)-E2 and E2-Ub(T)-E3) was imaged from two grids.<br>Biochemical analysis: Each condition was examined on at least two different days using three preparations of protein with three independent replicates. All attempts at replication were successful. All biochemical experiments were performed in triplicate, as indicated in the figure legends. For a detailed description, please see the 'Statistics and reproducibility' section in the methods. |

| Randomization | For cryo-EM 3D refinements, all particles are randomly split into two groups and two independent reconstructions are generated. The two groups are refined independently and the Fourier Shell Correlation (FSC) between the independent reconstructions is computed according to gold-standard procedure. Randomization is not applicable to other experiments described here. |
|---|---|
| Blinding | Blinding was not performed for cryo-EM image analysis as it requires manual evaluation at the step of image processing to ensure high-quality reconstructions. Blinding is not relevant to biochemical experiments because no group allocation was involved. Data were analyzed using unbiased methods. |

# Reporting for specific materials, systems and methods

We require information from authors about some types of materials, experimental systems and methods used in many studies. Here, indicate whether each material, system or method listed is relevant to your study. If you are not sure if a list item applies to your research, read the appropriate section before selecting a response.

## Materials & experimental systems

| n/a | Involved in the study |
|---|---|
| ☒ | Antibodies |
| ☒ | Eukaryotic cell lines |
| ☒ | Palaeontology and archaeology |
| ☒ | Animals and other organisms |
| ☒ | Clinical data |
| ☒ | Dual use research of concern |
| ☒ | Plants |

## Methods

| n/a | Involved in the study |
|---|---|
| ☒ | ChIP-seq |
| ☒ | Flow cytometry |
| ☒ | MRI-based neuroimaging |

## Plants

| Seed stocks | n/a |
|---|---|
| Novel plant genotypes | n/a |
| Authentication | n/a |

