## [Peer Review File · Nature]

Manuscript Title: Structural basis for transthiolation intermediates in the ubiquitin pathway

Reviewer Comments & Author Rebuttals

Reviewer Reports on the Initial Version:

Referees' comments:

Referee #2 (Remarks to the Author):

Conjugation of ubiquitin and ubiquitin-like proteins to substrates proceeds via a cascade of E1, E2 and E3 enzymes and involves at least one transthiolation step, from E1 to E2. Depending on the nature of the E3 this is followed by another transthiolation step to form an E3~Ub thioester intermediate, before the final Ub transfer onto the substrate. Transthiolation is an isoenergetic reaction and at present it is unclear which mechanisms impose directionality during this process. While previous studies reported structures of E1-E2 and ternary E2-Ub-E3 complexes to illuminate the ubiquitin transfer steps using a variety of crosslinking strategies to form the stable complexes, these studies suffered from non-native geometries of the transition state mimics used to trap these complexes, which were imposed by the type of chemistry used to trap the complexes.

Here, the authors have developed a novel bis-electrophilic ubiquitin probe that enables investigation of the transthiolation reaction by trapping two cysteines (E1/E2 or E2/E3 active sites) covalently while mimicking the tetrahedral transition state. Using this novel probe the authors analysed the E1-E2 and E2-E3 (HECT) transthiolation reactions in atomic detail by cryo-EM allowing them to capture a continuum of structures along the transthiolation pathway, and subsequently interrogated important contacts and conformational changes seen in these structures by biochemical experiments.

The approach presented here not only provides a new tool that will be of great interest to those studying transthiolation reactions, but also provides unprecedented detail of the conformational changes that take place during ubiquitin transfer from E1 to E2 and subsequently to a HECT E3 ligase.

Overall, the data presented are of high quality. However, it would be helpful if the authors were more explicit about the resolution limits of specific structures captured along the conformational continuum. The relevant data are all shown in supplementary info but any uncertainties imposed by resolution limits should be made clearer during the discussion of all the different states adopted in the main text, especially with respect to the positions of ubiquitin molecules, as this is crucial to assess the robustness and certainty of the conclusions drawn.

Specific points:

1) The first sentence of the Abstract seems rather disconnected from the core of this work which is

about transthiolation in the ubiquitin pathway.

2) The Methods are described in appropriate detail. However, the synthesis of the novel biselectrophilic probe is described in another manuscript. Is this accompanying manuscript describing PSAN probe synthesis likely to be published soon? This paper needs to be available alongside the current study to enable others to build on the work described here.

3) Line 134: The explanation why “The Ub(T) position.... differs by rotation of 55° from that reported in a crystal structure...linked to E2 Cdc34...” feels a bit hand-waving. Are there any differences between the E2s that could explain this observation? Any other possibilities?

4) The authors use doubly-loaded in different contexts: line 133: 1E1-2Ub “doubly Ub-loaded E1” and in Figure 2 “doubly-loaded E1” for the E1-Ub-E2 adduct. While I realize there is a distinction in the wording, it nevertheless is confusing for those not intimately familiar with the field. Consider being more specific.

5) It would be helpful if a cartoon was added to Figure 3 to illustrate the transfer model described on page 8, line 191 ff.

6) Figure 3: The EM densities and models shown in 3b and 3d are too small to see any relevant details.

Overall, figures of structural models (Fig 1c/d; 3c) are very small and lack sufficient labelling thereby making it difficult to distinguish different areas and follow the descriptions in the text.

7) Figure 4e and f: please label the model in panel e to indicate the crossover helix and position of the areas of interest magnified in panel f.

8) Overall, I would have liked to see a summary figure each for the transthiolation intermediates for the E1-Ub-E2 and E2-Ub-E3 (HECT) in which the structures presented in this study are compared to previous structures, which represent the “initial” and “final” states as determined by crystallography or cryo-EM.

For example, how much variation is seen in the position of ubiquitin in E2-Ub/E3 or E3-Ub across all available structures of these complexes, and how does this compare to the differences between the 7 states described here?

Referee #3 (Remarks to the Author):

In this manuscript the Lima lab addresses the fundamental mechanistic question how transthiolation can occur efficiently, while the reaction is isoenergetic. They study two separate reactions, transfer of ubiquitin (Ub) from E1 to E2 and transfer of Ub from E2 to HECT E3 ligases. This was enabled by a chemical approach that allowed stabilization of transition states in trans-thiolation with correct distances, published in a related manuscript. These were used in cryo-EM analyses to provide structural analysis of E1 to E2 as well as E2 to HECT E3 early transthiolation intermediates. The structures have decent resolution and reveal the existence of multiple intermediates.

Painstakingly detailed analysis of the various substates in the cryo-EM dataset provide detailed pictures of the transthiolation process. For the E1 to E2 transthiolation they separate complexes with one or two ubiquitins bound. They initially use focused classification of the doubly Ub-loaded states on the position of the Ub-thiol (Ub(T)) to ten classes and find that Ub(T) moves between a donor and acceptor position, but this does not seem to correlate to much change in other aspects of the complex. They then use 3DFlex to classify either the doubly or the single Ub loaded complex into five clusters, and see that only in the double-Ub loaded state, this reveals the hydrolysis and release of phosphate from the ATP site and correlates with the movement from the donor to the acceptor state, in line with the much faster reaction for the fully loaded state.

For the E2 to E3 transthiolation they also analyze different substates and realize that they capture early intermediates of the reaction, in contrast to an previous attempt that was more of a product complex. They show that the catalytically competent state is different than for RING E3 ligases and the relevance of this state is validated by mutagenesis. They hypothesize that it is this distinct binding that involves remodeled E2 loop residues between amino acids 86-92, and that it is their remodeling that prevents the backward reaction.

Using E2 mutant analysis they show that mutations that promote the E1 to E2 reaction, impede the E2 to E3 reaction, in line with the notion that the direction of the reactions is different with E2 changing from acceptor to donor, and the end-product must be protected.

Overall this is an impressive achievement. The analysis is very carefully done and meticulously described. The gels are unbelievably crisp, and the number of individual states shown is mind-boggling.

Main point:

For a general audience the current description is highly inaccessible:

- The abstract does not formulate a question, nor does it make clear what the structures actually show or how they are interpreted
- The legends of the figures do not guide the reader to understand why they are looking at all these details.
- The discussion has more clarity, but even here one has to search for the main supporting points that lead to these conclusions.

Other points

- Please show the U-net figure of the 3DFlex analysis to illustrate the scatter plots of the final distribution of learned particle latent coordinates and the route from cluster 1 to cluster 5.

Minor issues

- line 343: all sorts of steps are mentioned, but why not ATP hydrolysis?
- Line 413-416 the buffer during fluorescein labelling is not mentioned; same for line 444
- Line 469: please define the ice/salt bath
- Transthiolation assays: would be nice to define the scale of the reactions
- Fig 3b,d: there is a lot of information in these figures, and it is hard to find the main message; maybe one could consider not showing the (T) site here, as it does not change. Alternatively better legends could direct the reader to the loss of the phosphate density in d faster
- Fig 5c,d: it is not clear if the normalization is to individual WT or the mean of the 9 (or 18) replicates.

Textual

- Line 50 : very long sentence
- Line 401: 'applied to applied to' remove one applied to
- Line 424: supernatant lysate or lysate supernatant?
- Line 573 : s missing in statistics
- Page 56: extended data fig 12 there are two datasets 2, presumably one must be dataset 1?

Referee #4 (Remarks to the Author):

Kochanczyk et al have applied an elegant strategy to capture the transition states of transthioation intermediates in the ubiquitin pathway. Conjugation of ubiquitin to substrates is a three-step process involving an E1 activating enzyme, an E2 conjugating enzyme and an E3 ligase. The E1 first adenylates ubiquitin then forms a thioester bond with the C-terminus of ubiquitin. A second ubiquitin is then adenylated prior to transfer of the ubiquitin to the E2 via transthioation. The E2~ubiquitin thioester can then transfer ubiquitin directly to substrate with the help of a RING type E3 or can carry out transfer of ubiquitin to HECT, RBR or RCR type E3s again via transthioation. While a range of different approaches have been used previously to capture these transthioation intermediates none have accurately mimicked the tetrahedral geometry of the transition state. What the authors have done here is synthesise the bis-electrophilic probe 3-[phenylsulphonyl]-4-aminobut-2-enenitrile (PSAN) that captures the two cysteines involved in transthioation and the ubiquitin c-terminus in a tetrahedral state. The only difference from the native intermediate is that the C-terminal Ubiquitin Gly76 oxyanion is substituted by a cyanomethyl group. This intermediate is not completely stable, but removing the negative charge from the oxyanion stabilised the transition state long enough for analysis by single particle cryoEM. In both the E1-Ub-E2 and E2-Ub-E3 analyses what is revealed is the entire pathway of molecular transitions as the ubiquitin moves from E1 to E2 and from E2 to E3. While some of the E1-Ub-E2 and E2-Ub-E3 states have been captured previously using cruder approaches this is the first time that a continuum of structures has been captured that describe the entire process. The structures have been captured at high resolution and in cases where structural predictions have been tested, the biochemical approaches used have been entirely appropriate and this has validated the interactions. My only criticism is that I did not feel that the figures conveyed the considerable structural changes that were taking place. In figure 2 where the ubiquitin C-ter is covalently linked to the E1 and E2 active site thiols the body of ubiquitin moves between donor and acceptor positions through a 180 degree rotation and a 35Å translation. As the figure is so small (it will be even smaller in the journal) it is difficult to pick out these changes and it would be good if there was a larger more diagrammatic representation with arrows describing the rotation and translation.

Author Rebuttals to Initial Comments:

We thank the reviewers for their comments. We responded positively to each comment by altering the text and/or figures, a summary of which follows:

Referees' comments:

Referee #2 (Remarks to the Author):

Conjugation of ubiquitin and ubiquitin-like proteins to substrates proceeds via a cascade of E1, E2 and E3 enzymes and involves at least one transthiolation step, from E1 to E2. Depending on the nature of the E3 this is followed by another transthiolation step to form an E3~Ub thioester intermediate, before the final Ub transfer onto the substrate. Transthiolation is an isoenergetic reaction and at present it is unclear which mechanisms impose directionality during this process. While previous studies reported structures of E1-E2 and ternary E2-Ub-E3 complexes to illuminate the ubiquitin transfer steps using a variety of crosslinking strategies to form the stable complexes, these studies suffered from non-native geometries of the transition state mimics used to trap these complexes, which were imposed by the type of chemistry used to trap the complexes.

Here, the authors have developed a novel bis-electrophilic ubiquitin probe that enables investigation of the transthiolation reaction by trapping two cysteines (E1/E2 or E2/E3 active sites) covalently while mimicking the tetrahedral transition state. Using this novel probe the authors analysed the E1-E2 and E2-E3 (HECT) transthiolation reactions in atomic detail by cryo-EM allowing them to capture a continuum of structures along the transthiolation pathway, and subsequently interrogated important contacts and conformational changes seen in these structures by biochemical experiments.

The approach presented here not only provides a new tool that will be of great interest to those studying transthiolation reactions, but also provides unprecedented detail of the conformational changes that take place during ubiquitin transfer from E1 to E2 and subsequently to a HECT E3 ligase.

We thank the reviewer for the expert synopsis and interest.

Overall, the data presented are of high quality. However, it would be helpful if the authors were more explicit about the resolution limits of specific structures captured along the conformational continuum. The relevant data are all shown in supplementary info but any uncertainties imposed by resolution limits should be made clearer during the discussion of all the different states adopted in the main text, especially with respect to the positions of ubiquitin molecules, as this is crucial to assess the robustness and certainty of the conclusions drawn.

We added explicit references to the resolution or resolution range of structures as they are discussed in the main text, qualifying as necessary in cases where ubiquitin is more mobile.

Specific points:

1) The first sentence of the Abstract seems rather disconnected from the core of this work which is about transthioesterification in the ubiquitin pathway.

We appreciate this suggestion. The abstract has been modified to reflect the core of this work more accurately in response to this reviewer and reviewer 3.

2) The Methods are described in appropriate detail. However, the synthesis of the novel biselectrophilic probe is described in another manuscript. Is this accompanying manuscript describing PSAN probe synthesis likely to be published soon? This paper needs to be available alongside the current study to enable others to build on the work described here.

The accompanying manuscript was provided for reviewer consideration and was submitted at the same time as this manuscript. The manuscript describing PSAN probe synthesis has been published and is now referenced in the revised paper.

3) Line 134: The explanation why “The Ub(T) position.... differs by rotation of 55o from that reported in a crystal structure...linked to E2 Cdc34...” feels a bit hand-waving. Are there any differences between the E2s that could explain this observation? Any other possibilities?

We agree, our comments were diffuse because we were unable to pinpoint the precise reasons for why ubiquitin conformations differ. With that said, the conformation observed for Ub in complex with E1 (PDB 7k5j) with Ub linked to Cdc34 via a lysine in place of its active site cysteine is close to that observed in structures of isolated Cdc34 in complex with Ub that is not linked to E2 but bound to the small molecule CC0651 (PDB 4mdk). This conformation is also observed for Cdc34-Ub not bound to E1 with Ub linked to Cdc34 via a lysine in proximity of its active site cysteine (PDB 6nyo) also with CC0651. This suggests that Cdc34 binds to ubiquitin in this conformation regardless of a covalent linkage to Cdc34 or to E1, making Cdc34 a unique E2 with respect to its interactions with Ub. This interaction seems specific to Cdc34 as it is the basis of specificity of CC0651. As such, we removed our reference to the possible methodological differences and now reference these other PDBs as evidence that interactions may differ depending on E2s.

In response to this comment and those offered in point 8 below, we now include panels d and e in Extended Data Fig. 3 to compare structures presented in this study with previous structures including E1/Cdc34^{A141K}-Ub.

4) The authors use doubly-loaded in different contexts: line 133: 1E1-2Ub “doubly Ub-loaded E1” and in Figure2 “doubly-loaded E1” for the E1-Ub-E2 adduct. While I realize there is a distinction in the wording, it nevertheless is confusing for those not intimately familiar with the field. Consider being more specific.

We corrected the terminology throughout to state singly or doubly-loaded E1-Ub(T)-E2 and removed instances of ‘Ub-loaded’ to make it more consistent throughout the text and figures.

5) It would be helpful if a cartoon was added to Figure 3 to illustrate the transfer model described on page 8, line 191 ff.

A cartoon for the doubly loaded complex is shown in Fig 5e illustrating the transfer model proposed. This could be moved to Figure 3 but given other modifications to this Figure, additions to others, and space limitations we prefer to keep it in Figure 5. We also include additional reference to Supplementary Video 1 which captures these movements in a manner not possible in static figures.

6) Figure 3: The EM densities and models shown in 3b and 3d are too small to see any relevant details.

Overall, figures of structural models (Fig 1c/d; 3c) are very small and lack sufficient labelling thereby making it difficult to distinguish different areas and follow the descriptions in the text.

Fig 3c,e – We now show enlarged (A) site densities with labels by removing panels showing (T) site density as suggested by reviewer 3 as the (T) site densities remain unchanged and are shown in Extended Data Fig 5.

Fig 2c/d – We removed the top view of E1 complexes to add a panel showing and comparing Ub rotation/translations between states 1 and 10 (e). We also added panels showing models for ubiquitin in each of the 10 states per singly and doubly loaded E1s relative to the transthiolation active site in Extended Data Figure 3.

7) Figure 4e and f: please label the model in panel e to indicate the crossover helix and position of the areas of interest magnified in panel f.

Fig 4d,f (formerly e and f) – We added labels and rotated the orientation slightly to include the view from the back of the active site, now labeling and indicating the crossover helix and positions of areas of interest magnified in panel f. We added labels indicating the crossover helix.

8) Overall, I would have liked to see a summary figure each for the transthiolation intermediates for the E1-Ub-E2 and E2-Ub-E3 (HECT) in which the structures presented in this study are compared to previous structures, which represent the “initial” and “final” states as determined by crystallography or cryo-EM.

For example, how much variation is seen in the position of ubiquitin in E2-Ub/E3 or E3-Ub across all available structures of these complexes, and how does this compare to the differences between the 7 states described here?

We added additional figures comparing our structures with published structures for both E1-E2 and E2-E3 in Extended Data Figure 3 for E1 and Extended Data Figure 9 for E3.

Referee #3 (Remarks to the Author):

In this manuscript the Lima lab addresses the fundamental mechanistic question how transthiolation can occur efficiently, while the reaction is isoenergetic. They study two separate reactions, transfer of ubiquitin (Ub) from E1 to E2 and transfer of Ub from E2 to HECT E3 ligases. This was enabled by a chemical approach that allowed stabilization of transition states in trans-thiolation with correct distances, published in a related manuscript. These were used in cryo-EM analyses to provide structural analysis of E1 to E2 as well as E2 to HECT E3 early transthiolation intermediates. The structures have decent resolution and reveal the existence of multiple intermediates.

Painstakingly detailed analysis of the various substates in the cryo-EM dataset provide detailed pictures of the transthiolation process. For the E1 to E2 transthiolation they separate complexes with one or two ubiquitins bound. They initially use focused classification of the doubly Ub-loaded states on the position of the Ub-thiol (Ub(T)) to ten classes and find that Ub(T) moves between a donor and acceptor position, but this does not seem to correlate to much change in other aspects of the complex. They then use 3DFlex to classify either the doubly or the single Ub loaded complex into five clusters, and see that only in the double-Ub loaded state, this reveals the hydrolysis and release of phosphate from the ATP site and correlates with the movement from the donor to the acceptor state, in line with the much faster reaction for the fully loaded state.

For the E2 to E3 transthiolation they also analyze different substates and realize that they capture early intermediates of the reaction, in contrast to an previous attempt that was more of a product complex. They show that the catalytically competent state is different than for RING E3 ligases and the relevance of this state is validated by mutagenesis. They hypothesize that it is this distinct binding that involves remodeled E2 loop residues between amino acids 86-92, and that it is their remodeling that prevents the backward reaction.

Using E2 mutant analysis they show that mutations that promote the E1 to E2 reaction, impede the E2 to E3 reaction, in line with the notion that the direction of the reactions is different with E2 changing from acceptor to donor, and the end-product must be protected.

Overall this is an impressive achievement. The analysis is very carefully done and meticulously described. The gels are unbelievably crisp, and the number of individual states shown is mind-boggling.

Thank you for this expert synopsis and appreciation of the work.

Main point:

For a general audience the current description is highly inaccessible:

- The abstract does not formulate a question, nor does it make clear what the structures actually show or how they are interpreted

We appreciate this suggestion. The abstract has been modified to remove the first sentence (reviewer 2), and to pose a question and to better reflect the focus of this work more accurately.

- The legends of the figures do not guide the reader to understand why they are looking at all these details.

We revised figure legend titles to prepare the reader for what they will see and to improve clarity.

- The discussion has more clarity, but even here one has to search for the main supporting points that lead to these conclusions.

We made attempts to better highlight major points to improve impact and clarity.

Other points

- Please show the U-net figure of the 3DFlex analysis to illustrate the scatter plots of the final distribution of learned particle latent coordinates and the route from cluster 1 to cluster 5.

We added panels a-h to Extended Data Fig 5 to show particle latent coordinates from 3D variability analysis and their clustering.

Minor issues

- line 343: all sorts of steps are mentioned, but why not ATP hydrolysis?

ATP hydrolysis is not mentioned because it is not a step in this process, adenylation and pyrophosphate release are the steps in this reaction and the steps captured in our study, with release of AMP not mentioned as it takes place during thioester bond formation which is not encompassed in our study. This point is now moot because we revised this section to simply state that 'Unlike E1 to E2 transthiolation that is driven forward by coupling adenylation and transthiolation,'

- Line 413-416 the buffer during fluorescein labelling is not mentioned; same for line 444

We've added the requested information.

- Line 469: please define the ice/salt bath

Reference to the ice/salt bath is on line 439, we've added this information.

- Transthiolation assays: would be nice to define the scale of the reactions

We've added the scale of the reactions (presuming the reviewer is requesting the volumes for reactions).

- Fig 3b,d: there is a lot of information in these figures, and it is hard to find the main message; maybe one could consider not showing the (T) site here, as it does not change. Alternatively better legends could direct the reader to the loss of the phosphate density in d faster

This was addressed in part in response to reviewer #3 point 6.

Fig 3b,d – Clarity is improved by removing panels showing (T) site density, and enlarging panels showing (A) site density.

- Fig 5c,d: it is not clear if the normalization is to individual WT or the mean of the 9 (or 18) replicates.

Data are represented as relative to the mean of WT, this information has now been added.

Textual

- Line 50 : very long sentence

We revised the sentence to break it up to improve clarity.

- Line 401: ‘applied to applied to’ remove one applied to

Typo corrected.

- Line 424: supernatant lysate or lysate supernatant?

Typo corrected, to lysate supernatant.

- Line 573 : s missing in statistics

Typo corrected.

- Page 56: extended data fig 12 there are two datasets 2, presumably one must be dataset 1?

Typo corrected.

Referee #4 (Remarks to the Author):

Kochanczyk et al have applied an elegant strategy to capture the transition states of transthioation intermediates in the ubiquitin pathway. Conjugation of ubiquitin to substrates is a three-step process involving an E1 activating enzyme, an E2 conjugating enzyme and an E3 ligase. The E1 first adenylates ubiquitin then forms a thioester bond with the C-terminus of ubiquitin. A second ubiquitin is then adenylated prior to transfer of the ubiquitin to the E2 via transthioation. The E2~ubiquitin thioester can then transfer ubiquitin directly to substrate with the help of a RING type E3 or can carry out transfer of ubiquitin to HECT, RBR or RCR type E3s again via transthioation. While a range of different approaches have been used previously to capture these transthioation intermediates none have accurately mimicked the tetrahedral geometry of the transition state. What the authors have done here is synthesise the bis-electrophilic probe 3-[phenylsulphonyl]-4-aminobut-2-enenitrile (PSAN) that captures the two cysteines involved in transthioation and the ubiquitin c-terminus in a tetrahedral state. The only difference from the native intermediate is that the C-terminal Ubiquitin Gly76 oxyanion is substituted by a cyanomethyl group. This intermediate is not completely stable, but removing the

negative charge from the oxyanion stabilised the transition state long enough for analysis by single particle cryoEM. In both the E1-Ub-E2 and E2-Ub-E3 analyses what is revealed is the entire pathway of molecular transitions as the ubiquitin moves from E1 to E2 and from E2 to E3. While some of the E1-Ub-E2 and E2-Ub-E3 states have been captured previously using cruder approaches this is the first time that a continuum of structures has been captured that describe the entire process. The structures have been captured at high resolution and in cases where structural predictions have been tested, the biochemical approaches used have been entirely appropriate and this has validated the interactions.

Thank you for this expert synopsis and appreciation of the work.

My only criticism is that I did not feel that the figures conveyed the considerable structural changes that were taking place. In figure 2 where the ubiquitin C-ter is covalently linked to the E1 and E2 active site thiols the body of ubiquitin moves between donor and acceptor positions through a 180 degree rotation and a 35Å translation. As the figure is so small (it will be even smaller in the journal) it is difficult to pick out these changes and it would be good if there was a larger more diagrammatic representation with arrows describing the rotation and translation.

Fig 2c/d – We removed top views, and now include panels showing Ub rotations/translations in this Figure as well as in Extended Data Figures 3 and 9 in response to this reviewer and comments from reviewers 2 and 3.

Reviewer Reports on the First Revision:

Referee #2 comments:

Remarks to the Author:

The authors have made a number of changes to their manuscript to respond to the concerns raised by the reviewers, especially with respect to the Figures. This has significantly increased the clarity of the manuscript, and I'm now happy to recommend publication.

Reviewer Reports on the First Revision:

Referee #3 comments:

Remarks to the Author:

In this updated version, the authors have followed reviewer suggestions to the letter and this has resulted in one extra sentence in the abstract, shortened paragraph headers and a number of clearer figures with more accessible legends, besides some useful additions, such as very nicely done comparison to existing structures. This certainly has helped with accessibility of the paper. However, the main text was hardly changed, and consequently I'm still concerned that the result is difficult to understand for many Nature readers. Unfortunately, having read it now several times, I've also lost the ability to give further suggestions how to improve accessibility without losing content. Otherwise happy to see this impressive work published.

minor details :

- Legend fig 3g: Stimulation, not simulation i think
- Title Extended Data Fig. 4 Local resolution estimates for cryo-EM reconstructions please add which structures in title
- Extended Data Fig 6 and 7 are duplicated; probably intended Fig 7 is missing

Author Rebuttals to reviewer comments:

Referee #2 (Remarks to the Author):

The authors have made a number of changes to their manuscript to respond to the concerns raised by the reviewers, especially with respect to the Figures. This has significantly increased the clarity of the manuscript, and I'm now happy to recommend publication.

We thank the reviewer, no changes were required made in response to these comments.

Referee #3 (Remarks to the Author):

In this updated version, the authors have followed reviewer suggestions to the letter and this has resulted in one extra sentence in the abstract, shortened paragraph headers and a number of clearer figures with more accessible legends, besides some useful additions, such as very nicely done comparison to existing structures. This certainly has helped with accessibility of the paper. However, the main text was hardly changed, and consequently I'm still concerned that the result is difficult to understand for many Nature readers. Unfortunately, having read it now several times, i've also lost the ability to give further suggestions how to improve accessibility without losing content. Otherwise happy to see this impressive work published.

We thank the reviewer, no further changes were made in response to these comments, we agree, all further attempts to generalize resulted in lost content.

minor details :

- Legend fig 3g: Stimulation, not simulation i think

Typo corrected, thank you.

- Title Extended Data Fig. 4 Local resolution estimates for cryo-EM reconstructions
please add which structures in title

Structures explicitly added in the title, corrected, thank you.

- Extended Data Fig 6 and 7 are duplicated; probably intended Fig 7 is missing

We double and triple checked these files on our computers and on the Nature website (uploaded files) and the reviewer is mistaken, extended data Fig 6 and 7 are not duplicated although we imagine that similar layouts led to this perception.